# Visualizing synaptic dopamine efflux with a 2D composite nanofilm

Chandima Bulumulla[1], Andrew T Krasley[1], Ben Cristofori-Armstrong[1,2],
William C Valinsky[1], Deepika Walpita[1], David Ackerman[1], David E Clapham[1],
Abraham G Beyene[1]*

[1]Janelia Research Campus, Howard Hughes Medical Institute, Ashburn, United
States; [2]Center for Advanced Imaging, The University of Queensland, Queensland,
Australia

**Abstract** Chemical neurotransmission constitutes one of the fundamental modalities of communication between neurons. Monitoring release of these chemicals has traditionally been difficult to carry out at spatial and temporal scales relevant to neuron function. To understand chemical neurotransmission more fully, we need to improve the spatial and temporal resolutions of measurements for neurotransmitter release. To address this, we engineered a chemi-sensitive, two-dimensional composite nanofilm that facilitates visualization of the release and diffusion of the neurochemical dopamine with synaptic resolution, quantal sensitivity, and simultaneously from hundreds of release sites. Using this technology, we were able to monitor the spatiotemporal dynamics of dopamine release in dendritic processes, a poorly understood phenomenon. We found that dopamine release is broadcast from a subset of dendritic processes as hotspots that have a mean spatial spread of ≈ 3.2 μm (full width at half maximum [FWHM]) and are observed with a mean spatial frequency of one hotspot per ≈ 7.5 μm of dendritic length. Major dendrites of dopamine neurons and fine dendritic processes, as well as dendritic arbors and dendrites with no apparent varicose morphology participated in dopamine release. Remarkably, these release hotspots co-localized with Bassoon, suggesting that Bassoon may contribute to organizing active zones in dendrites, similar to its role in axon terminals.

*For correspondence:
beyenea@janelia.hhmi.org

Competing interest: The authors declare that no competing interests exist.

## Editor's evaluation

This is a very exciting study that presents a novel approach to examining dopamine release with spatial precision that is so far unrivaled. This manuscript is also important and timely in the field of biosensor development and of potential interest to neuroscientists who study neurochemical release. It introduces a synthetic nanofilm with high spatiotemporal resolution and quantal sensitivity to dopamine measurement. By utilizing this technology to visualize sub-cellular dopamine efflux, the work provides new insights into the spatiotemporal dynamics and protein machinery of somatodendritic dopamine release. The authors identify hotspots for DA release and also provide evidence for DA release in the presence of TTX, suggesting the occurrence of quantal release.

## Introduction

Chemical neurotransmission generally falls under one of two broad categories: fast synaptic transmission or neuromodulation. Synapses that mediate rapid communication between most excitatory and inhibitory synapses in the brain primarily employ glutamate or γ-aminobutyric acid (GABA). Such chemical communication occurs at highly specialized synaptic structures that have nanoscale spatial organization and operate with millisecond temporal precision (*Clements et al., 1992*; *Südhof, 2012*;

**eLife digest** To form the vast and complex network necessary for an organism to sense and react to the world, neurons must connect at highly specialized junctions. Individual cells communicate at these 'synapses' by releasing chemical signals (or neurotransmitters) such as dopamine, a molecule involved in learning and motivation.

Despite the central role that synapses play in the brain, it remains challenging to measure exactly where neurotransmitters are released and how far they travel from their release site. Currently, most tools available to scientists only allow bulk measurements of neurotransmitter release.

To tackle this limitation, Bulumulla et al. developed a new way to measure neurotransmitter release from neurons, harnessing a technique which uses fluorescent nanosensors that glow brighter when exposed to dopamine. These sensors form a very thin film upon which neurons can grow; when the cells release dopamine, the sensors 'light up' as they encounter the molecule. Dubbed DopaFilm, the technology reveals exactly where the neurotransmitter comes from and how it spreads between cells in real time. In particular, the approach showed that dopamine emerges from 'hot spots' at specific sites in cells; it also helped Bulumulla et al. study how dopamine is released from subcellular compartments that have previously not been well characterized.

Improving the sensors so that the film could detect other neurotransmitters besides dopamine would broaden the use of this approach. In the future, combining this technology with other types of imaging should enable studies of individual synapses with intricate detail.

*Choquet and Triller, 2013*). In contrast, neuromodulators, including biogenic amines, neuropeptides, and hormones, operate at different spatiotemporal scales. Neuromodulatory synapses do not exhibit a close apposition to their partners but act on receptors that are extrasynaptically localized, and signal through G-protein coupled intracellular mechanisms (*Greengard, 2001*). This suggests that neuromodulators diffuse from their release sites in adequate quantities to influence target receptors (*Agnati et al., 1995*).

Generally, the nature of the chemical synapse is less well understood for neuromodulators, which differ from their classical counterparts not only in their mechanisms of action on receptors but also in their secretory apparatus and spatiotemporal dynamics. For dopamine, one of the most important neuromodulators in the brain, the challenge is compounded by certain unique features pertinent to dopamine neurobiology. Dopaminergic neurons are known for their large size and exhibit highly ramified axonal arborizations and dense varicosities (*Matsuda et al., 2009*; *Pacelli et al., 2015*; *Bolam and Pissadaki, 2012*). Studies have sought to establish the molecular determinants of dopamine release from these dense axonal arbors, but these attempts do not rely on measurement of dopamine efflux with single release site resolution (*Ducrot et al., 2021*; *Liu et al., 2018*). Previous studies have demonstrated that dopamine neurons possess the machinery for co-release of other neurotransmitters, including glutamate and GABA, but to what extent, if any, co-release events spatially overlap remains insufficiently understood (*Stuber et al., 2010*; *Sulzer et al., 1998*; *Tritsch et al., 2016*; *Tritsch et al., 2012*; *Fortin et al., 2019*). Electrochemical and microdialysis assays in midbrain regions have shown that somatodendritic release of dopamine constitutes an important component of dopamine signaling, with notable implications in disease and behavior (*Beckstead et al., 2004*; *Gantz et al., 2013*; *Björklund and Lindvall, 1975*; *Cheramy et al., 1981*; *Cragg et al., 1997*; *Ludwig et al., 2016*). However, the spatial and temporal dynamics of dendritic release events and their regulatory mechanisms remain poorly characterized. In sum, there is a pressing need for tools with appropriate sensitivity, kinetics, and subcellular spatial resolution to explore mechanisms of neurochemical release, including that of dopamine.

In this work, we developed an assay that facilitates visualization of the efflux of dopamine from active zones with synaptic resolution and quantal sensitivity. Here, we define efflux to mean the two-dimensional (2D) broadening of released dopamine quanta. In axonal arbors, we observed that dopamine release arises from a sparse set of varicosities, and we were able to assign the observed spatially defined effluxes to individually identified boutons. In dendrites and dendritic arbors of dopamine neurons, we similarly visualized spatially resolved effluxes of dopamine from putative dendritic active zones, which have been less well understood than release zones in axons. Our results show

that dopamine neurons can sustain robust levels of release from their dendritic processes. Indeed, we demonstrate that dendritic release is fast and Ca$^{2+}$ dependent, suggesting gating of dopamine release by a Ca$^{2+}$-mediated release machinery, reminiscent of classical active zones. Dopamine efflux was observed at major dendrites, fine dendritic processes, and at junctions of the soma and dendrites but rarely directly from the cell body itself. Soma of dopamine neurons were observed to receive strong dopaminergic input due to release and diffusion from proximal dendrites. Retrospective super-resolution imaging at identified release sites shows that Bassoon, long established as a presynaptic scaffolding protein of the cytomatrix in presynaptic active zones, is also enriched at the vicinity of dopamine release sites in dendrites and dendritic arbors. The expression of vesicular SNARE protein synaptobrevin-2 correlated with dopamine release activity in dendritic segments, establishing its functional utility in dopamine neurons. Therefore, active zones in dendrites appear to utilize a constellation of presynaptic and SNARE-complex proteins that are responsible for coordinating release, similar to those observed in classical synapses. This study offers a technology that facilitates visualization of the spatial and temporal efflux of dopamine and deploys the technology to shed light on somatodendritic dopamine release, a facet of dopaminergic neurobiology that has been insufficiently characterized.

## Results

### Visualizing active zone dopamine efflux from axons and dendrites

To a first-order approximation, once released from a synaptic active zone, the temporal evolution of the released chemical should approximate that of diffusion from a point source, characterized by an isotropic expansion from the point of release but constrained by transporter activity and local three-dimensional ultrastructure. Such a signal can only be fully measured if the sensing platform sufficiently samples the underlying signal in both the spatial and temporal domains. However, the inability of current technologies to measure chemical efflux sufficiently in the spatial domain limits our ability to study chemical synapses. We addressed this challenge by using near-infrared (NIR) fluorescent dopamine nanosensors to image single dopamine release sites from rat primary midbrain neuronal cultures. The nanosensors are assembled from oligonucleotide-functionalized, single-wall carbon nanotubes in solution phase (*Beyene et al., 2018*; *Kruss et al., 2014*) and have previously been used to image dopamine release in striatal acute slices and cultured cells but were lacking in synaptic information (*Beyene et al., 2019*; *Kruss et al., 2017*). In this study, we drop-cast glass coverslips with dopamine nanosensors to produce a 2D layer of a turn-on fluorescent, dopamine-sensitive surface that can effectively image dopamine diffusion from a point source (*Figure 1A*). Temporally, the sensors exhibit subsecond turn-on responses, which enabled real-time imaging of dopamine's temporal evolution. We named the engineered surface as DopaFilm, a 2D engineered film that affords video-rate filming of dopamine spatiotemporal dynamics with subcellular spatial and millisecond temporal resolution.

DopaFilm was validated by co-culturing rat primary midbrain dopamine neurons with cortical and hippocampal neurons on the nanofilm surface for up to 6 weeks (typical period for an experiment) (*Figure 1B and C*). Dopamine neurons in culture exhibited stereotyped morphology, with thick major dendrites arising from the soma that go on to ramify into dendritic arbors, and axonal processes that branch and arborize on a scale of millimeters from a single neuron (*Figure 1—figure supplement 1*). The electrophysiological properties of hippocampal neurons grown on DopaFilm were similar to neurons that were grown on sensor-free substrates, which suggested that the electrophysiological phenotype of neurons grown DopaFilm was not altered (*Figure 1—figure supplement 2*). DopaFilm fluoresced in the NIR to short-wave infrared (SWIR) regions of the spectrum (850–1350 nm) when imaged with a 785 nm excitation laser, permitting its multiplexed deployment with existing optical technologies with no spectral overlap (*Figure 1—figure supplement 3*). The surface exhibited an isotropic turn-on response when exposed to exogenous dopamine wash, suggesting uniform sensor coverage and response (*Figure 1—figure supplement 3A*). Dopamine wash experiments revealed DopaFilm has the sensitivity to detect 1 nM concentrations, remained stable over the duration of a typical experiment, and has an apparent dissociation constant ($K_d$) of 268 nM (*Figure 1—figure supplement 3B-C*). This compares to half maximal effective concentrations(EC$_{50}$ )values of ~ 1 μM for D$_1$-like and ~ 10 nM for D$_2$-like dopamine receptors, suggesting that DopaFilm is sensitive to chemical secretions that have physiological relevance (*Rice and Cragg, 2008*). In order to utilize DopaFilm's advantageous photophysical properties, we developed a custom microscope that is optimized

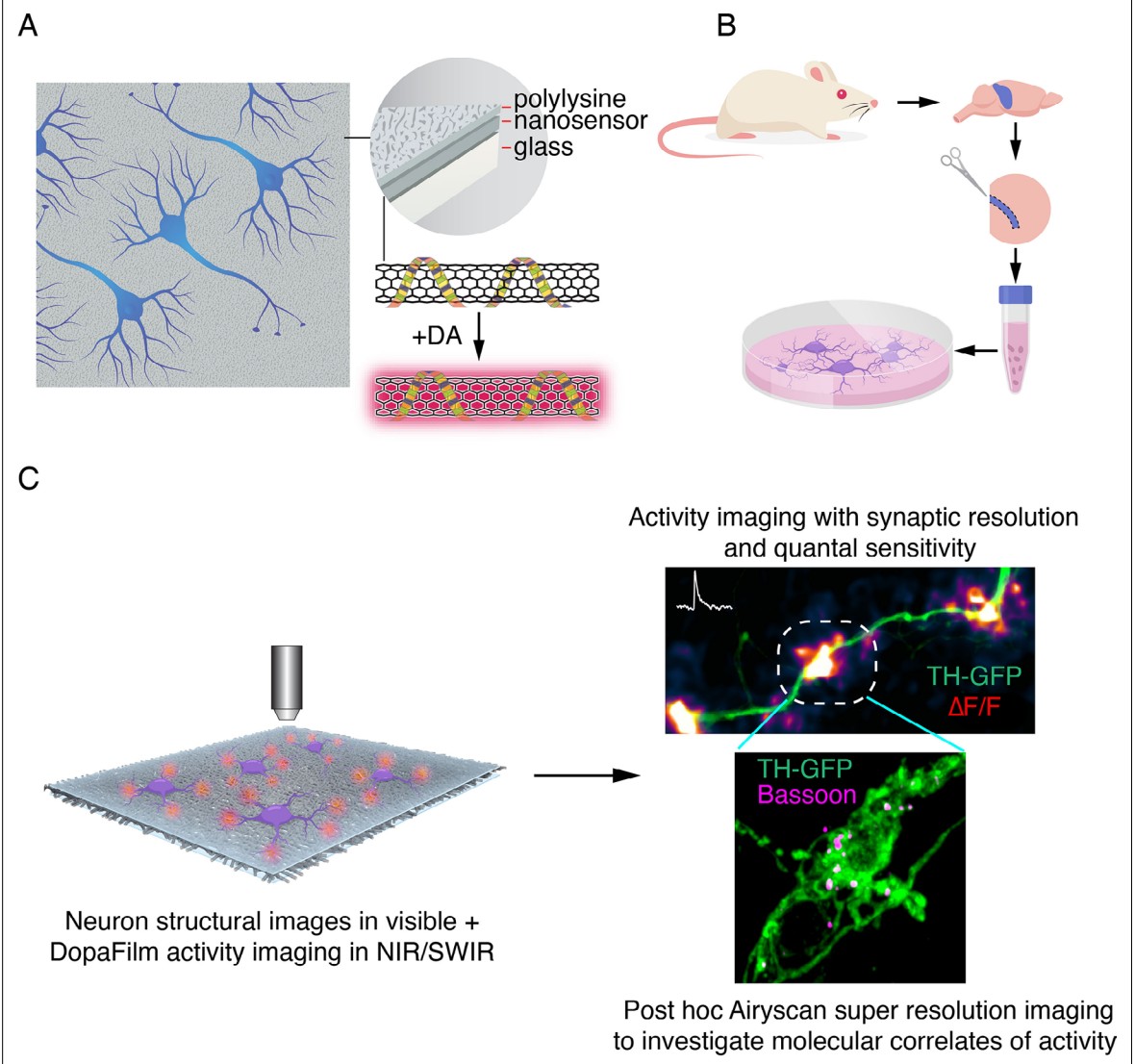

**Figure 1.** Schematic of DopaFilm imaging protocol. (**A**) Schematic of DopaFilm. (**B**) Workflow for preparing dopamine neuron primary cultures from the rat midbrain regions highlighted in blue. Neurons are grown on dishes with an engineered, chemi-sensitive, and fluorescent surface (DopaFilm) between polylysine and glass surfaces. (**C**) Imaging DopaFilm fluorescence activity in cultured dopamine neurons. Immunocytochemistry and Airyscan super-resolution imaging are carried out following DopaFilm activity imaging.

The online version of this article includes the following figure supplement(s) for figure 1:

**Figure supplement 1.** Tyrosine hydroxylase (TH) immunofluorescence of dopamine neurons grown on DopaFilm.

**Figure supplement 2.** Electrophysiology of hippocampal neurons cultured on a poly-D-lysine-only substrate (PDL) and DopaFilm.

**Figure supplement 3.** DopaFilm and solution-phase nanosensor characterization.

for broad spectrum imaging in the visible, NIR and SWIR regions of the spectrum (400–1400 nm), with integrated widefield and laser scanning confocal capabilities. The optimizations in the NIR and SWIR regions facilitated imaging and recording of activity with exceedingly high signal-to-noise (SNR) ratios, attaining SNRs in the range of 5–50 for most experiments. In this study, most imaging experiments were carried out in widefield epifluorescence mode using a 40×/0.8 NA objective (N40X-NIR, Nikon). This gave us a field of view (FOV) of 180 × 230 μm. In axonal arbors, the FOV contained several hundred dopaminergic varicosities, whereas in cell body regions, we could simultaneously image activity around the soma, major dendrites, and dendritic arbors. Post hoc immunofluorescence super-resolution images were collected on Zeiss LSM 880 with Airyscan mode.

We asked if dopamine neurons grown on DopaFilm can be evoked to release dopamine, and whether DopaFilm fluorescence transients can recapitulate the predicted spatiotemporal evolution of dopamine efflux. To drive this effort, we virally co-expressed TH-GFP (GFP expressed under the control of the rat tyrosine hydroxylase [TH] promoter) and Syn-ChrimsonR-tdTomato (the red shifted opsin, ChrimsonR, expressed under the control of synapsin promoter and fused to tdTomato for visualization). This co-expression paradigm facilitated identification and optical stimulation of putative dopamine neurons. About 79% of TH-GFP+ neurons were confirmed to be dopaminergic in retrospective immunofluorescence against TH, the rate-limiting enzyme in dopamine biosynthesis. On the other hand, all of the TH-immunoreactive neurons expressed the virally delivered TH-GFP transgene. During seeding, we optimized density of cells on DopaFilm such that mature neurons formed a monolayer of cells on the substrate. The seeding density allowed us to record activities arising from isolated dopamine neurons where no other neurons in the vicinity of the neuron of interest were TH+, ensuring that detected activity can be assigned to single identifiable processes with minimal crosstalk. We used 561 nm LED (5 pulses and 25 Hz) to depolarize dopaminergic neurons. A subset of dopamine neurons exhibited spontaneous spiking activity, and we applied no external stimuli in those cases. Furthermore, most dopamine neurons exhibited stochastic, temporally uncorrelated release events that appeared to be action potential independent, and those events were also included in our data. All imaging experiments were carried out 3–4 weeks postviral infection, and neurons were 4–5 weeks in vitro at the time of activity imaging.

We first imaged in axonal arbors, where dopamine release is relatively better characterized through microdialysis and voltammetry measurements (*Liu et al., 2018*; *Garris et al., 1994*; *Robinson et al., 2003*; *Robertson et al., 1991*). Whenever targeted to TH+ neurons (further confirmed by retrospective immunofluorescence experiments), our stimulation protocol elicited robust fluorescence transients from DopaFilm (*Figure 2A and B*, *Figure 2—animation 1*), and the fluorescence hotspots co-localized with TH+ boutons in axonal arbors (*Figure 2C*). We observed diffusive broadening of the fluorescence hotspots in subsequent imaging frames (*Figure 2A*, +1 s), and fluorescence transients returned to baseline in the poststimulation epoch (*Figure 2A* post, *Figure 2B*). The observed spatiotemporal evolution of DopaFilm hotspots is consistent with that of release and diffusion from multiple point-like sources localized in a 2D plane, with estimated mass diffusivities of $\approx 1.1 \pm 0.8 \times 10^{-6}$ cm$^2$ s$^{-1}$ (mean ± SD), in reasonable agreement with estimated values of diffusion coefficient for dopamine (*Cragg et al., 2001*). Despite the high density of axonal varicosities in the FOV, fluorescence transients were observed to emanate from a subset of varicosities while another subset of varicosities produced no corresponding ΔF/F fluorescence hotspots (*Figure 2C*, red arrows). The percentage of release-competent boutons varied greatly in axonal arbors, ranging from 5% in some FOVs to 65% in others, with a mean of 32% of putative boutons participating in release. This sparse-release observation is in agreement with results from previous studies (*Liu et al., 2018*; *Pereira et al., 2016*). DopaFilm hotspot activities were also observed to co-localize with dendrites of dopamine neurons (*Figure 2D and E*) and were additionally noted to arise from dendritic processes that commingled with the soma (*Figure 2F and G*). The turn-on and clearance kinetics of the measured transients in axons were 0.46 ± 0.16 s (mean ± SD) for time to peak ($\tau_{peak}$) and 3.83 ± 0.8 s (mean ± SD) for first-order decay time constant ($\tau_{off}$) (*Figure 2—figure supplement 1*). The turn-on kinetics is slower than those reported for the genetically encoded dopamine sensors GRAB$_{DA}$ ($\approx 100$ ms) and dLight (reported as $\tau_{1/2}$ of $\approx 10$ ms followed by a plateau of $\approx 100$ ms) (*Patriarchi et al., 2018*; *Sun et al., 2018*). On the other hand, decay kinetics appears to be slower than dLight (reported as $\tau_{1/2} \approx 100$ ms) and comparable to or faster than those reported for GRAB$_{DA}$ ($\approx 3$–17 s for variants). For comparison, G-protein-gated inwardly rectifying K$^+$ (GIRK) current-based dopamine dynamics measurements exhibited $\tau_{peak} \approx 250$ ms, whereas carbon fiber recordings peaked in $\tau_{peak} \approx 300$ ms (*Marcott et al., 2014*). This suggests that the kinetic properties of DopaFilm transients are comparable with the range of reported values from existing tools. The DopaFilm was responsive only to the activity of TH+ neurons. Stimulation of ChrimsonR+ in TH− neurons did not elicit fluorescence transients in DopaFilm, and neither evoked nor spontaneous activities were noted in neurons that lacked TH immunoreactivity despite being TH-GFP+ in the viral expression paradigm (*Figure 2—figure supplement 2*). Both evoked and spontaneous DopaFilm fluorescence transients were absent when imaging in extracellular 'Ca$^{2+}$-free' media (*Figure 2—figure supplement 3*).

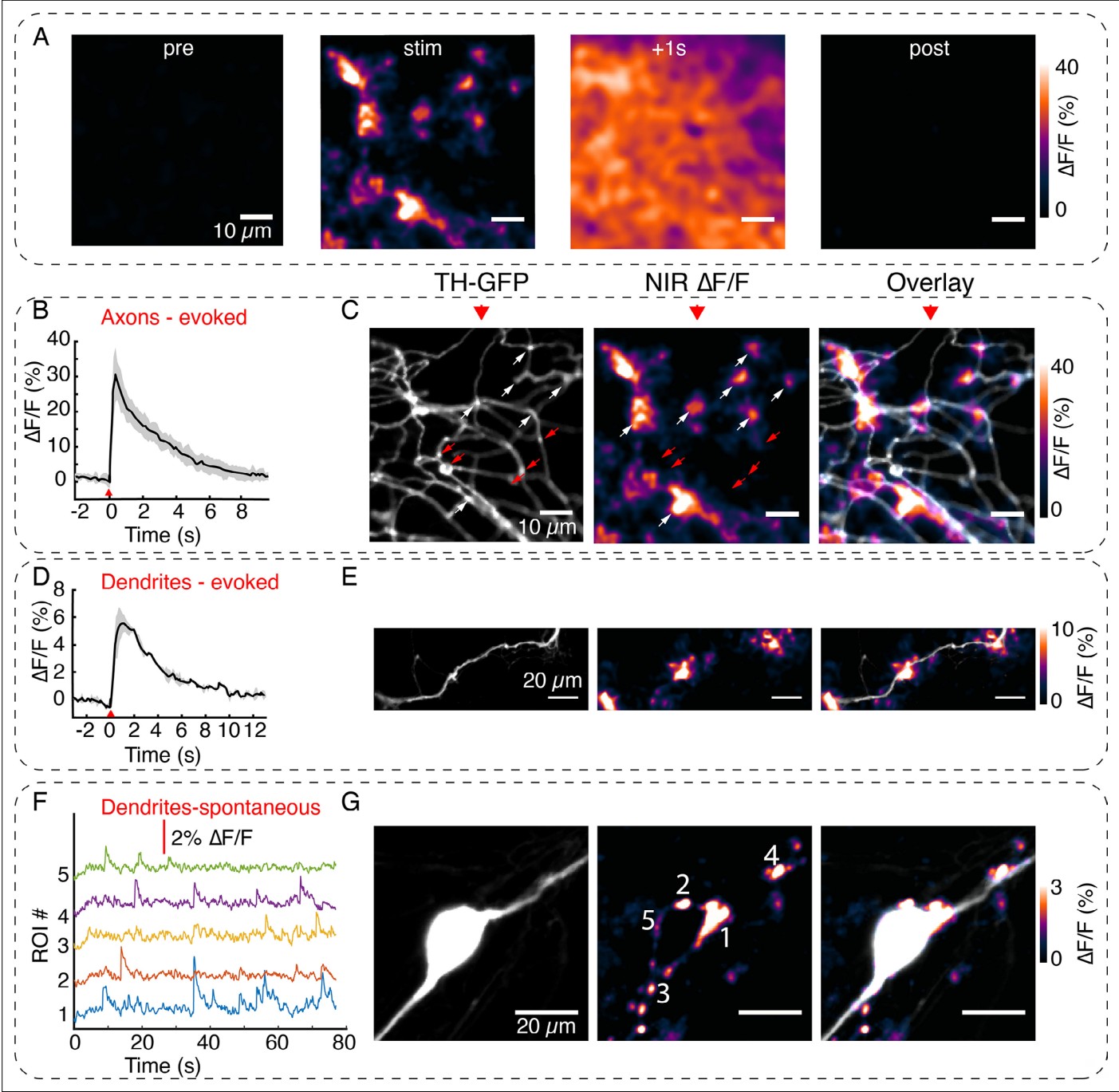

**Figure 2.** Visualizing active zone dopamine efflux from axons and dendrites. (**A**) ΔF/F images before stimulation (pre), at the time of stimulation (stim), 1 s after stimulation (+1 s) and after return to baseline (post). (**B**) Mean ± SD of ΔF/F traces from the imaging field of view in (**A**) averaged over n=5 repeat stimulation runs. (**C**) Tyrosine hydroxylase-GFP (TH-GFP) image of axonal arbor and near-infrared (NIR) ΔF/F image shown in (**A**) and overlay. The NIR ΔF/F frame corresponds to 'stim' and before diffusive broadening of the hotspots. (**D, E**) TH-GFP and ΔF/F activity from a dendrite of a dopamine neuron and overlay. Cell body not shown. Activity traces from dendrite are averaged over n=3 stimulations. (**F, G**) Spontaneous activity from dendrites around cell body of a dopamine neuron and maximum intensity projection of the ΔF/F stack and overlay. (**F**) Shows ΔF/F activity traces from regions of interest (ROI) numbered in (**G**). Red wedges in (**B**) and (**D**) = time of optical stimuli.

The online version of this article includes the following video and figure supplement(s) for figure 2:

**Figure supplement 1.** Turn-on and clearance kinetics of DopaFilm in axonal and dendritic processes.

**Figure supplement 2.** Tyrosine hydroxylase (TH) immunofluorescence is used to verify identity of putative dopamine neurons.

*Figure 2 continued on next page*

Figure 2 continued

**Figure supplement 3.** DopaFilm activity imaging in normal imaging buffer (artificial cerebrospinal fluid [ACSF] with 2 mM Ca$^{2+}$) and buffer with no extracellular Ca$^{2+}$ (Ca$^{2+}$-free ACSF).

**Figure 2—animation 1.** Imaging in axons.

## DopaFilm hotspots localize to defined varicosities

We next explored the consistency of DopaFilm ΔF/F hotspot dynamics over multiple stimulation epochs and asked if repeat stimulations in the same FOV generated DopaFilm hotspots that localize to the same set of boutons. We carried out imaging in an FOV where we applied the same optical stimuli to drive hotspot activity over separate imaging sessions. In axon terminals, we observed that repeat stimulations can be carried out with rest periods of ~ 2–3 min between stimuli, giving rise to a consistent set of DopaFilm hotspot activities, and suggesting that release-competent dopamine neuron boutons likely have a high probability of release (*Figure 3A*, *Figure 3—figure supplement 1A-B*). To evaluate the spatial specificity of the observed hotspots, we computed the intensity-weighted centroid of each DopaFilm hotspot and compared the centroids across multiple stimulation repeats. We found that hotspot centroids were remarkably consistent across repeat stimulations (*Figure 3C*). To determine if the hotspots localized to the same set of varicosities, we compared DopaFilm hotspot centroids with the centroid of a TH+ varicosity that spatially overlapped with the DopaFilm hotspot from the overlay image (*Figure 3B*). We defined dopamine varicosities as puncta where TH-GFP intensity is at least 3 × the mean intensity of TH-GFP expression along the process and computed the centroids of these TH+ boutons (*Figure 3B* center, green puncta, *Figure 3D*). Using an arbitrarily chosen origin as a reference point (0,0), we compared centroids of DopaFilm hotspots and TH+ boutons. Our results showed that ΔF/F hotspot centroids and bouton centroids matched with remarkable consistency across stimuli (no offset in some regions of interest [ROIs], <5 camera pixels for all ROIs, equivalent to <1.7 µm; *Figure 3E*). These experiments demonstrate that DopaFilm activity hotspots can be faithfully localized to the same set of boutons across stimulation epochs and are therefore likely driven by the efflux of dopamine from putative active zones of these boutons. The ability to localize synaptic dopamine efflux to specific boutons is unique to this study and to the best of our knowledge has not been demonstrated before.

It is notable that not all TH-GFP+ boutons produced a corresponding DopaFilm activity (*Figure 2C*, *Figure 3B*, red arrows indicate no activity) despite appearing to satisfy the morphological criteria for a varicosity. We considered the possibility that the failure to detect activity from some putatively silent boutons was the result of rapid dopamine clearance, before DopaFilm detection. To test if dopamine clearance was critical to 'silent' boutons, we applied saturating levels of nomifensine (NOM; 10 µM), a dopamine-specific reuptake inhibitor ($K_i \approx$ 100 nM). Application of NOM altered the clearance kinetics of DopaFilm transients, defined by a first-order decay time constant, $\tau_{off}$ (mean ± SD of $\tau_{off}$ for artificial cerebrospinal fluid [ACSF]: 4.38 ± 0.84 and for +NOM: 5.70 ± 1.47, p-value=3×10$^{-4}$, *Figure 3—figure supplement 1E*) and dramatically affected the overall kinetic profile of ΔF/F traces, indicated by mean FWHM of 1.85 s before and 4.45 s after application of NOM (*Figure 3F–J*). Additionally, the measured peak amplitude of the ΔF/F trace was higher after application of NOM (mean peak ΔF/F = 15.7% before and 24.5% after application of NOM; *Figure 3F–J*). Intriguingly, NOM altered the turn-on profile of DopaFilm transients (*Figure 3—figure supplement 1C-D*). When imaging under normal ACSF, the clearance profile of dopamine followed an approximately monotonic decay that is punctuated by brief instances of upward deflection (blue arrows in *Figure 3—figure supplement 1C*). We posit that these are dopamine release events whose signal is arrested by dopamine transporters (DATs). Saturation of DATs by NOM permits these release events in the poststimulation epoch to be detected by DopaFilm. This suggests that DATs may therefore play a role in arresting spill-over of dopamine during a train of release events. Importantly, application of NOM did not reveal any subthreshold activity at silent varicosities that could have gone undetected in pre-NOM imaging sessions (*Figure 3F and G*). Thus, we conclude that DopaFilm hotspot activity arises from dopamine release at varicosities, and that the absence of DopaFilm fluorescence transient is likely an indication of a lack of dopamine release at release-incompetent varicosities. When coupled with the fact that DopaFilm activity is not observed in TH− cells, and that activity is absent when imaging in Ca$^{2+}$-free media, we conclude that DopaFilm ΔF/F activity is a consequence of release of endogenous

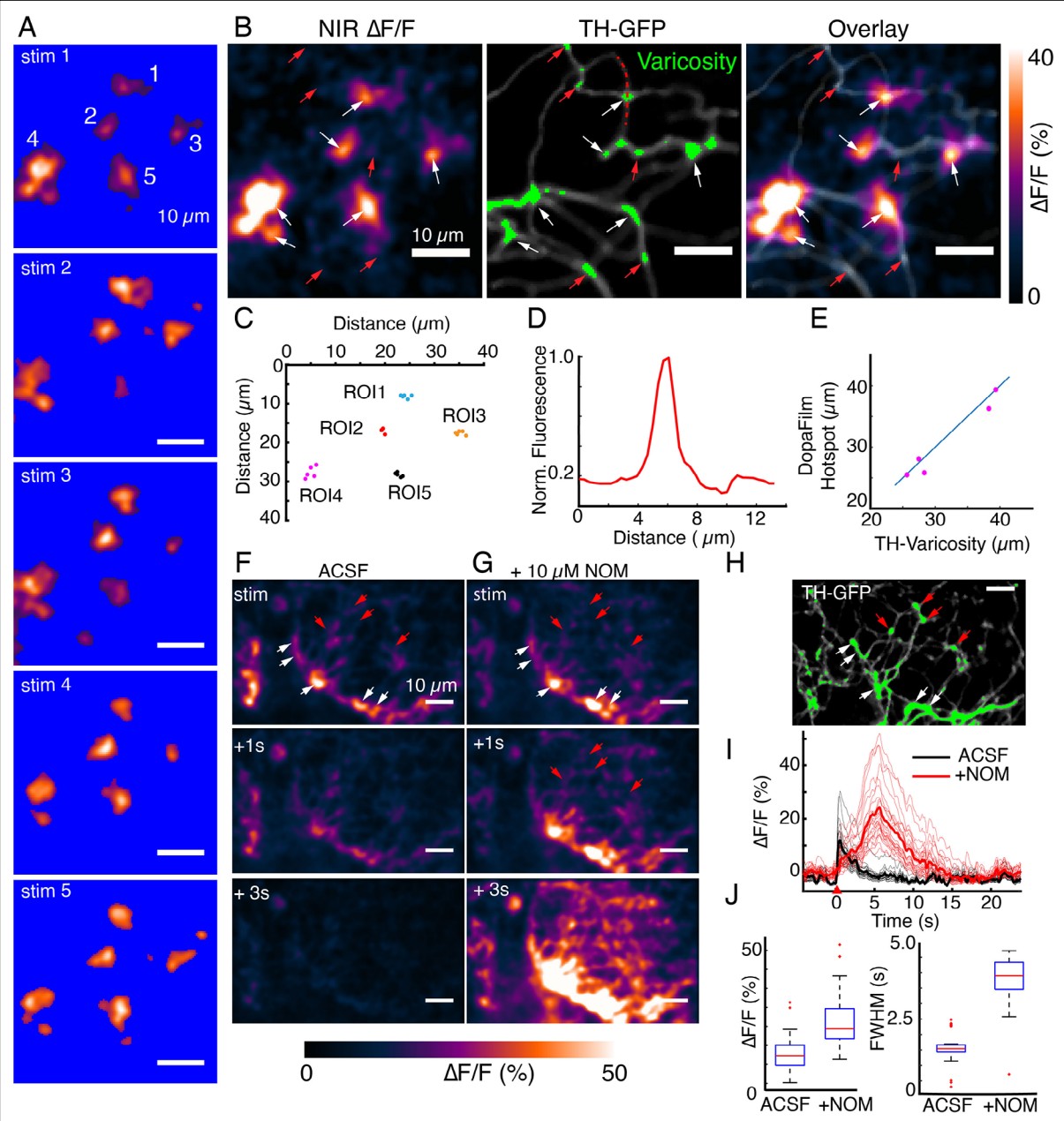

**Figure 3.** DopaFilm hotspots localize to defined varicosities. (**A**) Repeat optical stimulation produced a consistent set of DopaFilm hotspots (n=5 of n=10 stimulations shown), thresholded for better visualization. (**B**) Hotspot colocalization with tyrosine hydroxylase-GFP (TH-GFP) varicosities (green) for stim #1. Green colors indicate GFP mean intensity >3 × the mean intensity along the process. (**C**) Centroid of hotpots (regions of interest [ROIs] 1–5). Each data point of an ROI corresponds to one evoked imaging run and there are five total per ROI. (**D**) Line profile for one of the varicosities is shown in (**B**). Line profile is calculated along the red curve shown in (**B**). (**E**) Parity plot for TH and DopaFilm hotspot centroids. Centroid of TH varicosities and averaged centroid of DopaFilm hotspots are plotted. Distance is calculated from the top left corner (=origin) of near-infrared (NIR) ΔF/F image shown in (B). (**F, G**) Spatiotemporal dynamics of imaging in artificial cerebrospinal fluid (ACSF) and ACSF + nomifensine (+10 µM NOM), respectively. (**H**) TH-GFP image of field of view in (**F, G**). Red arrows = no release. Scale bar = 10 µm. (**I**) ΔF/F traces of hotspots in ACSF (black) and ACSF + 10 µM NOM (red) and their mean traces in bold. (**J**) Box plots comparing the effect of NOM on peak ΔF/F (left) and full width at half maximum (FWHM; right) of traces shown in (**I**). Unpaired t-test, $p < 10^{-4}$ for both ΔF/F and FWHM data. See Methods for box plot definitions.

The online version of this article includes the following figure supplement(s) for figure 3:

**Figure supplement 1.** (**A–B**) Robustness of hotspot activity during repeat stimulation in axons and (**C–E**): effect of nomifensine (NOM) on turn-on and clearance kinetics.

**Figure supplement 2.** Effect of glutamate transmission on dopamine release in dopaminergic and glutamatergic neuron co-culture system.

dopamine from active release sites. The NOM manipulation of the temporal dynamics of dopamine release provides additional evidence that DopaFilm possesses the kinetics necessary to recapitulate the dynamic behavior of dopamine release, diffusion, and clearance.

In our study, dopamine neurons are co-cultured with cortico-hippocampal neurons, and we explored if glutamatergic activity from neurons in co-culture could influence dopamine release. To investigate this, we carried out experiments in which α-amino-3-hydroxy-5-methyl-4-isoxazolepropionic acid (AMPA)-type glutamate receptor antagonist 2,3-dioxo-6-nitro-7-sulfamoyl-benzo[f]quinoxaline (NBQX) and N-methyl-D-aspartate (NMDA)-type glutamate receptor antagonist D-2-amino-5-phosphonovalerate (D-AP5) were bath-applied to the co-culture system while imaging release from dopamine neurons. We first examined neurons from which DopaFilm activity can be detected from spontaneous spiking events in which we applied no external stimulus to generate activity. We imaged from these neurons under ACSF (our normal imaging buffer) and then bath-applied NBQX (10 µM). DopaFilm activities that were detected before application of NBQX were absent in the postdrug imaging sessions (*Figure 3—figure supplement 2A-B*). Application of NBQX was sufficient to abolish these activities. Additionally, we examined the extent to which glutamatergic currents contributed to dopamine neuron depolarization during evoked activity imaging. To investigate this, we carried out imaging before and after glutamate receptor blockade with a combined application of NBQX and D-AP5. Here, such treatment resulted in reduced dopamine release as measured by the peak amplitude of ΔF/F traces and the area under the curve of ΔF/F traces (*Figure 3—figure supplement 2C-E*). In sum, these results indicate that DopaFilm offers an opportunity for direct measurement of dopamine release under pharmacological perturbations and suggests that our in vitro culture system may permit simplified explorations of local chemical circuitries that control dopamine release in the absence of complex circuit effects that may be encountered in vivo.

## DopaFilm detects quantal release of dopamine

We next sought to establish the limit of detection of DopaFilm. In in vitro experiments, we determined that DopaFilm exhibits high sensitivity to dopamine and can detect 1 nM concentrations in bath application experiments (*Figure 1—figure supplement 3B*). This suggested that DopaFilm may be sensitive enough to detect single events of quantized dopamine efflux from release sites. To determine the limit of detection from a practical sense, we carried out imaging experiments in a field of an axonal arbor of a dopamine neuron before and after bath application of tetrodotoxin (TTX), which inhibits action potential driven, synchronous neurotransmitter release while sparing stochastic and spontaneous release events. We first imaged activity in ACSF, our normal imaging buffer and then applied TTX. As expected, bath application of 10 µM TTX abolished synchronous, evoked release of dopamine but stochastic, temporally uncorrelated fluorescence transients, which we refer to as spontaneous release events, persisted (*Figure 4A*, *Figure 4—figure supplement 1*, *Figure 4—animation 1* before TTX, *Figure 4—animation 2* after TTX). The spatial extent of DopaFilm fluorescence hotspots was diminished after application of TTX (*Figure 4A, B and D*, *Figure 4—figure supplement 1A*, *Figure 4—figure supplement 2*), and the peak amplitude of transients was smaller compared to evoked release (*Figure 4E*). The TTX ΔF/F peaks were comparable to spontaneous activity peaks detected before TTX addition (peak ΔF/F (%) [mean ± SD]: 5.3 ± 1.6 for spontaneous activity vs 7.9 ± 4.6 for TTX) but exhibited different kinetic characteristics as measured by FWHM (s) (mean ± SD: 1.5 ± 1.0 for spontaneous activity vs 0.74 ± 0.70 for TTX, $p<10^{-4}$ in unpaired t-test) (*Figure 4E*). The decay time constants ($\tau_{off}$) for TTX transients were smaller than those of evoked releases (mean ± SD (s): 3.83 ± 0.8 vs 1.52 ± 0.53, *Figure 4—figure supplement 1E*). To evaluate the statistical spread of the observed $\tau_{off}$ values, we created a cumulative frequency distribution of $\tau_{off}$ values with and without TTX (for evoked release) and defined a parameter, $\tau_{10/90}$, as the *range* of $\tau_{off}$ for values that fall in the 0.1–0.9 quantile window in the cumulative distribution curve (*Figure 4—figure supplement 1F*). In axon terminals without TTX (for evoked release), we determined that $\tau_{10/90} \approx 1.95$ s, whereas $\tau_{10/90}$ for TTX data was ≈ 1.15 s, suggesting that TTX events have kinetic properties that are more similar to each other. Interestingly, we observed that TTX transients occurred with a frequency of (mean ± SD) 0.29 ± 0.24 $s^{-1}$ per release site, which contrasted with a frequency of (mean ± SD) 0.16 ± 0.13 $s^{-1}$ per release site for spontaneous activities before TTX addition (*Figure 4—figure supplement 1H-I*). Notably, DopaFilm fluorescence transients in TTX were still detected with a robust SNR of more than 5, where the SNR is defined as the ratio of ΔF to the SD of $F_0$ (*Figure 4C*, *Figure 4—figure supplement 3A*).

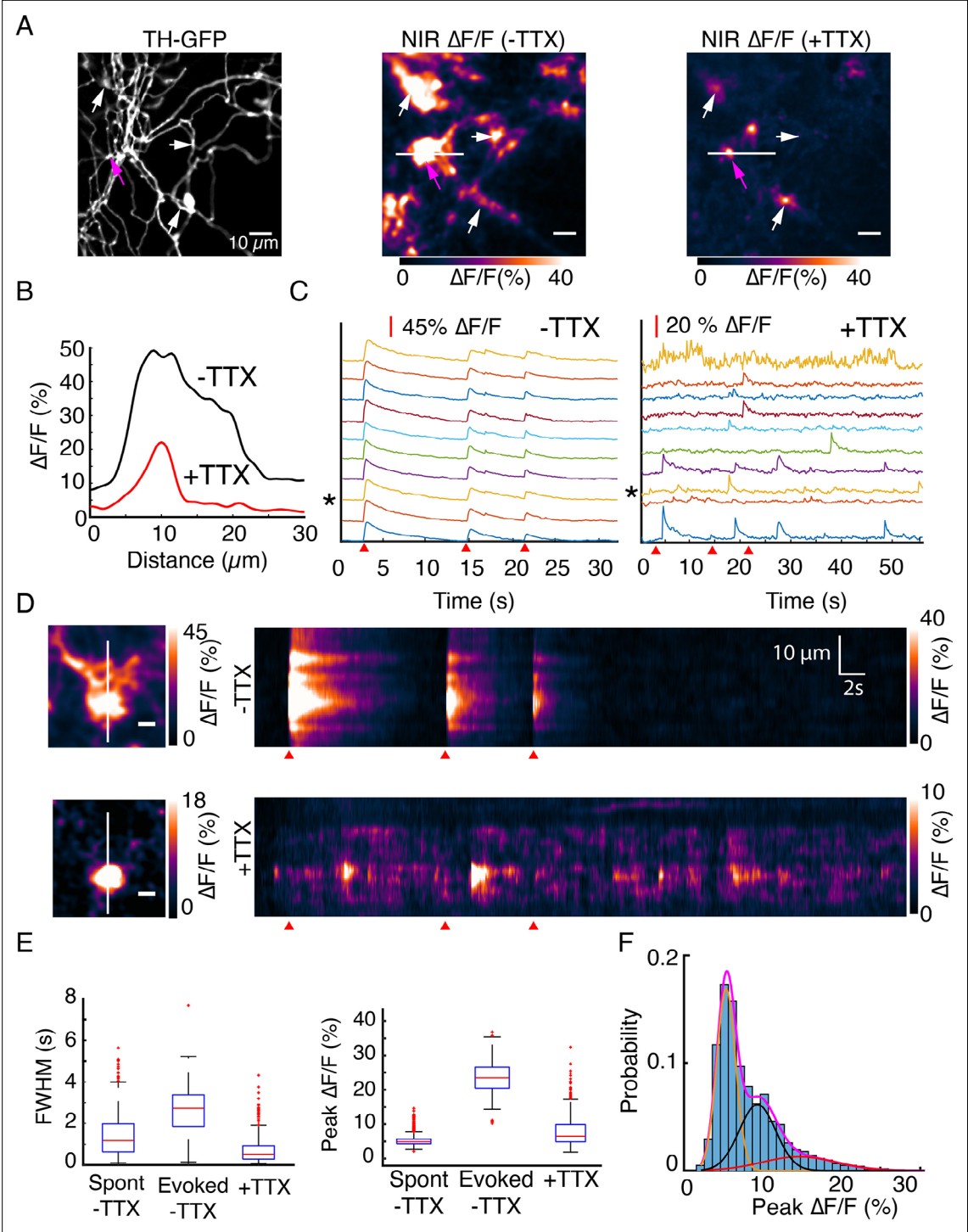

**Figure 4.** DopaFilm detects quantal release of dopamine. (**A**) Tyrosine hydroxylase-GFP (TH-GFP) image from an axonal arbor of a dopamine neuron (soma not shown). Peak ΔF/F of DopaFilm fluorescence transients with no tetrodotoxin (TTX) in imaging media (–TTX) and max ΔF/F projection of stack with 10 μM TTX in media (+TTX). Arrows provided to aid comparison across field of views. (**B**) ΔF/F profiles across the white lines depicted in (**A**). Line profiles correspond to the peak ΔF/F frames and are not averaged across stimuli. (**C**) Fluorescence transient traces for hotspots with (–TTX) and (+TTX), respectively. * indicates trace corresponding to magenta arrow in (**A**). Red wedges in (**C**) and (**D**) indicate times of optical stimuli. (**D**) Kymographs along the white lines shown in (**A**). Scale bar = 5 μm. (**E**) Box plots of full width at half maximum (FWHM) (s) and ΔF/F amplitude (%) for (–TTX, spontaneous and evoked) and (+TTX) cases. The term 'spont' refers to spontaneous activity before TTX addition, and 'evoked' refers to action potential driven synchronous release before TTX addition. FWHM (s) (mean ± SD): 1.5 ± 1.0 for 'spont –TTX', 2.6 ± 1.2 for 'evoked –TTX', and 0.74 ± 0.70 for '+TTX'. Peak ΔF/F (%) (mean ± SD): 5.3 ± 1.6 for 'spont –TTX', 23.7 ± 4.6 for 'evoked –TTX', and 7.9 ± 4.6 for '+TTX'. p<10⁻⁴ in unpaired t-test of the 'evoked –

*Figure 4 continued on next page*

*Figure 4 continued*

TTX' against non-evoked ('spon –TTX' and '+TTX') datasets. See Methods for box plot definitions. (**F**) Frequency histogram of ΔF/F amplitudes for TTX data shown in (**C**) and Gaussian fits to the experimental data. Magenta = sum of three Gaussian components shown in orange, black, and red. Means: $\mu_1$=5.7%, $\mu_2$=9.6%, and $\mu_3$=15.1%.

The online version of this article includes the following video and figure supplement(s) for figure 4:

**Figure supplement 1.** Application of tetrodotoxin (TTX) abolishes evoked release but spares spontaneous activities.

**Figure supplement 2.** DopaFilm activity imaging in an axonal arbor in: (**A**) normal artificial cerebrospinal fluid (ACSF) (**B**) ACSF with 10 μM of tetrodotoxin (TTX).

**Figure supplement 3.** Statistical model fits for tetrodotoxin (TTX) data.

**Figure 4—animation 1.** Imaging in axons before application of tetrodotoxin.

**Figure 4—animation 2.** Imaging in axons after application of tetrodotoxin.

---

Incidentally, the SNR for most of our TTX-free, optical stimulus-evoked release in axons frequently exceeded 30 (*Figure 4—figure supplement 1C-D*). In dense axonal arbors, where DopaFilm fluorescence hotspots could not be assigned to specific varicosities, application of TTX allowed visualization of spatial evolution from putative single release sites by eliminating release from neighboring active zones (*Figure 4A*, *Figure 4—figure supplement 1A*, *Figure 4—figure supplement 2*).

We pooled peak ΔF/F values obtained from TTX hotspot traces across the FOV of imaging and generated a histogram distribution of the peak ΔF/F values (*Figure 4—figure supplement 3A-B*). Inspection of single traces and the pooled histogram data suggested that DopaFilm transients in TTX may represent quantal events of different sizes or integer multiples of the same quantal size. To rationalize our experimental observation, we explored the suitability of statistical model fits for our experimental data. We first considered the use of Gaussian mixture models (GMM), which are often employed in quantal analysis of synaptic transmission events. Using Akaike information criteria (AIC) as a discriminating score, we evaluated the relative qualities of GMMs composed of one to four components. We found that a three-component GMM whose mean parameters (μ) are $\mu_1$=5.7%, $\mu_2$=9.6%, and $\mu_3$=15.1% ΔF/F offered the best fit (*Figure 4F*, *Figure 4—figure supplement 3C*). The spacing of the mean parameters, μ, suggested that DopaFilm fluorescence transients in TTX are likely driven by a quantized biological process where each upward deflection in DopaFilm fluorescence represented some integer multiple of a unitary fusion event. A four-component GMM returned a higher AIC score than the three-component AIC, suggesting overfitting (*Figure 4—figure supplement 3C*). We considered the possibility that our experimental data may simply be better described by a positively skewed distribution function instead of a GMM. To evaluate this possibility, we compared our chosen three-component GMM with a lognormal distribution. Lognormal distributions are positively skewed and are frequently encountered in biological processes (*Koch, 1966*). A single lognormal distribution is a two-parameter statistical model (μ and σ) and would therefore be treated favorably in AIC scoring compared to an eight-parameter three-component GMM (three pairs of μ and σ and two mixing probability parameters, π). The AIC rewards a model's predictive power, while penalizing the number of parameters used. In this way, multiple models fitting the same data can be compared. Here, we saw that the GMM's AIC score was better than that of the lognormal model (*Figure 4—figure supplement 3D-E*), suggesting that GMM explained the observed experimental variance better than a simple skewed distribution. In addition to AIC, we employed a graphical model discrimination method called quantile-quantile plot, which also supported our conclusion that GMM is a better statistical model for the experimental data (*Figure 4—figure supplement 3F-G*). In summary, our experimental results and statistical analysis demonstrate that DopaFilm transients that are measured in TTX represent quantal release events of dopamine and, to our knowledge, offer the only experimental observation of putative single-vesicle fusion events in which the released neurochemical can be visualized in both spatial and temporal domains.

## DopaFilm detects dopamine release from dendritic processes

To further demonstrate the utility of DopaFilm, we deployed it to study a poorly understood phenomena in dopamine neurobiology. Previous studies have demonstrated that dopamine is released from somatodendritic processes of dopamine neurons in the midbrain (*Beckstead et al., 2004*; *Gantz et al., 2013*; *Björklund and Lindvall, 1975*; *Cheramy et al., 1981*; *Cragg et al., 1997*;

Ludwig et al., 2016). Despite decades of research, however, the release of dopamine at somata and dendrites of dopamine neurons remains incompletely understood (Rice and Patel, 2015). Because microdialysis and electrochemical assays, which have historically been employed to measure dopamine, do not have subcellular spatial resolution, it is still not clear if dopamine release detected in perikarya of dopamine neurons emanates from soma or dendrites. This has necessitated the use of the term 'somatodendritic release' (Ludwig et al., 2016). Absence of small synaptic vesicles and synaptic morphologies in ultrastructural studies of dopamine neuron dendritic arbors has made the functional role of dendritic projections in midbrain regions less obvious (Nirenberg et al., 1996; Wassef et al., 1981; Wilson et al., 1977). We reasoned that DopaFilm could unravel some of these mysteries owing to its subcellular spatial and millisecond temporal resolution, and quantal sensitivity.

When deployed for imaging activity in somatodendritic compartments of dopamine neurons, DopaFilm detected both evoked and spontaneous transients that clearly emanated from MAP2+ and TH+ dendritic processes of dopamine neurons (Figure 5, Figure 5—figure supplement 1). MAP2 is enriched in dendrites and is absent in axons (Kosik and Finch, 1987); therefore, MAP2 immunoreactivity was used to distinguish between the dendritic and axonal nature of observed transients. DopaFilm transients were observed at dendritic varicosities (i.e. bouton-like structures on dendrites; Figure 5B) and dendritic processes that did not appear to have any varicose morphology (Figure 5—figure supplement 2, Figure 5—figure supplement 3). From a spatial perspective, release at dendrites often resembled the localized, hotspot-like activities seen when imaging in axonal arbors (Figure 5, Figure 5—figure supplement 2, Figure 6, Figure 5—animation 1 for evoked, Figure 5—animation 2 for spontaneous). In some dendritic processes, we observed activities that emanate from clustered active zones in contiguous segments of dendrites, producing hotspots that spread along the entire profile of the dendritic process (Figure 5—figure supplement 3A-B, Figure 5—animation 2). To rule out the possibility that what we perceived to be dendritic release could arise from commingling axonal processes in the same dendritic arbor, we amplified the TH-GFP signal with an anti-GFP antibody and performed post hoc Airyscan super-resolution imaging for all our images. We observed no axons that tracked with the dendrites and DopaFilm transients clearly localized to MAP2+ and TH-GFP+ processes, confirming their dendritic nature (Figure 5C and E, Figure 5—figure supplement 1, Figure 5—figure supplement 3B, Figure 6H). Activities from dendrites were detected reliably during repeat stimulation experiments across imaging sessions (Figure 5—figure supplement 4A-B). DopaFilm ΔF/F peak amplitudes and temporal FWHM were smaller in dendrites, but other kinetic properties such as turn-on ($\tau_{peak}$) and clearance ($\tau_{off}$) were similar when compared to transients in axonal release sites (Figure 5—figure supplement 4C-F). We observed a notable difference in the spatial propagation propensity of dopamine release in axonal arbors vs dendrites. In dendritic processes, DopaFilm hotspot activity remained confined to the vicinity of the release site, which contrasted with the more diffusive dynamics observed in axons (Figure 5—figure supplement 5). We computed the FWHM of each hotspot as a proxy to estimate the spatial spread of dopamine from a dendritic release site and obtained values of ≈ 3.2 µm ±3 µm (mean ± SD). In axons, we observed spatial spreads of ≈ 6.6 µm ± 3.6 µm (mean ± SD) (Figure 5—figure supplement 5C). Dendritic hotspots were encountered at a frequency of ≈ 7.5 µm ± 0.7 µm (mean ± SD) along the active dendrites which participated in release (Figure 5—figure supplement 6).

Despite the noted differences between axonal and dendritic release, our experiments showed that DopaFilm transients at dendritic processes shared many key features with those in axons. Fluorescence turn-on was fast for both evoked and spontaneous activities (Figure 5—figure supplements 1–3, Figure 5—figure supplement 7), suggesting availability of fusion-ready docked vesicles reminiscent of vesicular release at axonal terminals. Moreover, dendrites could sustain robust levels of activity in extended imaging experiments, suggesting an available pool of vesicles that can be quickly recruited, docked, and primed (Figure 5—figure supplement 4A-B, Figure 5—figure supplement 7). These data suggested that release of dopamine in dendrites may be mediated by molecular machinery comparable to those in axon terminals.

Moreover, we explored to what extent, if any, somata of dopamine neurons contribute to somatodendritic release. As was often the case in most imaging sessions, we saw little direct activity arising from soma of dopamine neurons (Figure 5A). When DopaFilm activity was noted near soma, such activity often fell into one of two categories. First, we noticed that DopaFilm activity can often be observed on the major dendrites of dopamine neurons (i.e. the stereotypic thick dendritic trunks that

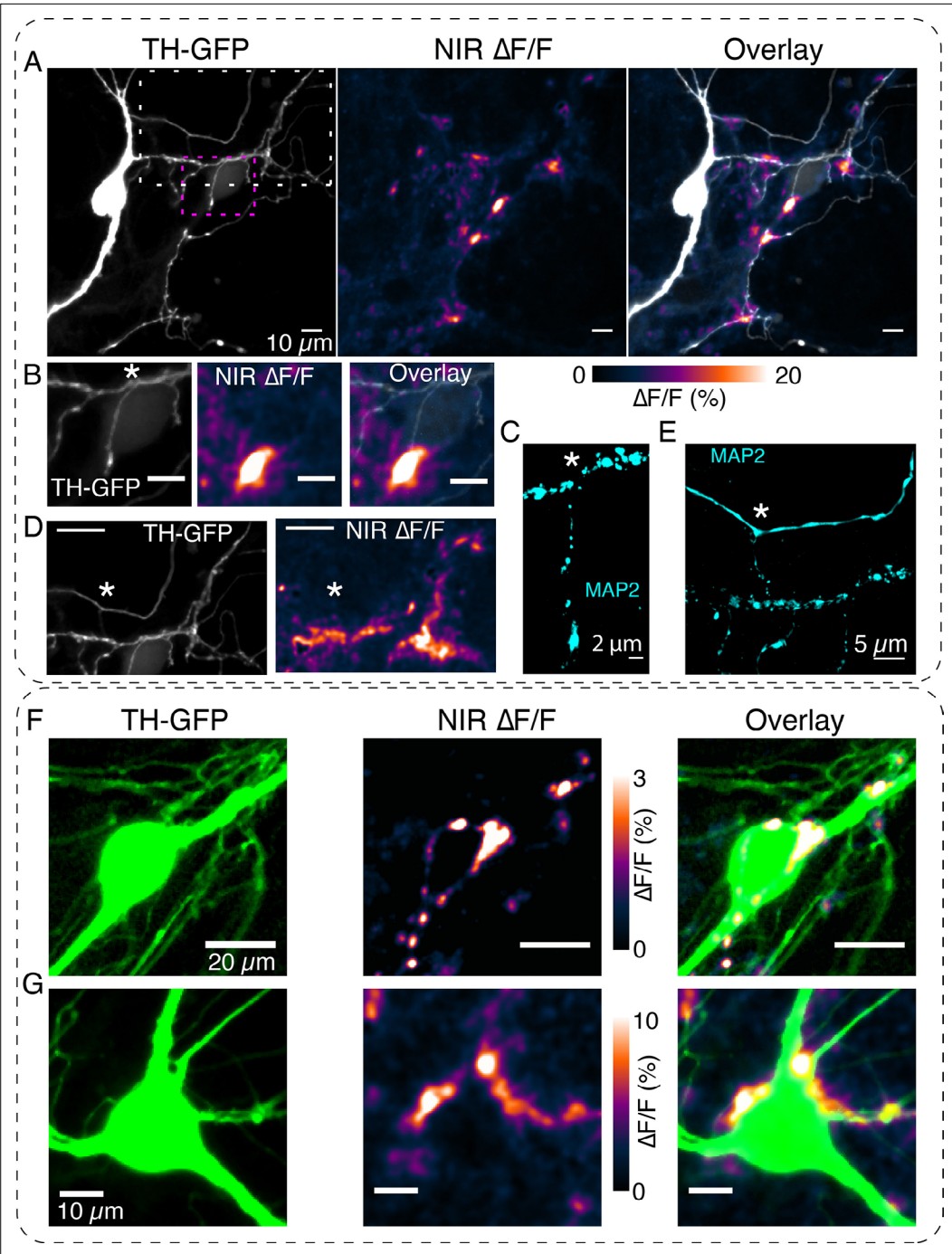

**Figure 5.** DopaFilm detects dopamine release activity from dendritic processes. (**A**) Tyrosine hydroxylase-GFP (TH-GFP) image of a dopamine neuron and its spontaneous DopaFilm activity. Maximum intensity projection from a ΔF/F stack is shown, overlayed with TH-GFP. (**B**) Area bounded by magenta box in (**A**) and close-up of its spontaneous DopaFilm activity and overlay. Scale bar = 10 μm. (**C**) MAP2 signal corresponding to (**B**). Use * to compare field of views (FOVs) in (**B**) and (**C**). (**D**) Area bounded by white box in (**A**) and close-up of its spontaneous DopaFilm activity. Scale bar = 20 μm. See TH-GFP and near-infrared (NIR) ΔF/F overlay image in *Figure 5—figure supplement 1*. (**E**) MAP2 signal corresponding to (**D**). Use * to compare FOVs in (**D**) and (**E**). (**F, G**) DopaFilm activity arising from dendrites that commingle with the soma of dopamine neurons. Notice activity from fine dendritic processes that appear in proximity to the soma. Transient traces for activity in (**F**) are shown in *Figure 2F*.

The online version of this article includes the following video and figure supplement(s) for figure 5:

**Figure supplement 1.** MAP2+ immunofluorescence demonstrates dendritic nature of processes.

*Figure 5 continued*

**Figure supplement 2.** Spontaneous DopaFilm activity imaging at a dendrite of a dopamine neuron.

**Figure supplement 3.** DopaFilm activity imaging of a dopamine neuron.

**Figure supplement 4.** (**A-B**) Robustness of hotspot activity during repeat stimulation in dendrites and (**C-E**) Comparison of spatiotemporal dynamics in axons and dendrites.

**Figure supplement 5.** Comparison of spatiotemporal dynamics of DopaFilm hotspots in axonal arbors and dendritic processes after optical stimulation (evoked).

**Figure supplement 6.** DopaFilm hotspot activity along dendritic processes.

**Figure supplement 7.** Imaging in dendritic processes of a dopamine neuron.

**Figure 5—animation 1.** Evoked release from a dendritic process.

**Figure 5—animation 2.** Spontaneous activity from a dendritic process.

**Figure 5—animation 3.** Dendritic activity around soma of dopamine neuron.

depart from the soma), including at the junction between the soma and dendrite (*Figure 5—figure supplement 2*, *Figure 6—figure supplement 2*). Second, we visualized release from dendrites that commingled with the cell body, often in such proximity that it appears as if activity arose from the soma (*Figure 5F–G*, *Figure 5—figure supplement 2*, *Figure 5—animation 3*). However, the subcellular imaging capability of DopaFilm clarifies, even in such cases, that activity was likely driven by dendrites that appeared to innervate the soma.

## DopaFilm enables interrogation of the molecular correlates of dopamine release

We next used DopaFilm to examine the protein machinery involved in organizing release in dendritic processes. In classical synapses, release of neurotransmitters from axon terminals occurs at a highly specialized structure called the synaptic active zone. The release site is enriched in large scaffolding proteins (Bassoon and Piccolo) and active zone protein complexes that dock and prime synaptic vesicles (RIM, RIM-BP, ELKS, and MUNC-13) as well as SNARE-complex proteins (VAMP, SNAP-25, and syntaxin) that carry out vesicle fusion in the final exocytotic step (*Südhof, 2012*). We wanted to know if DopaFilm dendritic activity was co-located with the expression of proteins that are classically involved in neurochemical release at axon terminals. To explore this question, we examined Bassoon, a large scaffolding protein that is classically used as a marker for presynaptic terminals in axons (*Gundelfinger et al., 2015*), and whose disruption leads to loss of synaptic transmission (*Altrock et al., 2003*). We first performed DopaFilm activity imaging in dendritic processes of dopamine neurons. We then immunostained for Bassoon and performed post hoc Airyscan imaging at dendritic locations that exhibited hotspots of DopaFilm activity. These experiments showed that Bassoon was enriched at dendritic locations where we observe DopaFilm ΔF/F hotspot activity (*Figure 6*, *Figure 5—figure supplements 2–3*). In some images, DopaFilm activity hotspots were localized directly to a Bassoon punctum (*Figure 6B and C*, *Figure 5—figure supplement 2*), while in others, we observed that DopaFilm activity co-localized with enriched clusters of Bassoon puncta (*Figure 6E and F*). Additionally, when DopaFilm activity was observed along a whole segment of a dendritic process (as opposed to a localized hotspot on the process), we noted that Bassoon is likewise enriched along the process (*Figure 6G, H and I*). Importantly, density of Bassoon expression at a location correlated positively with the magnitude of ΔF/F activity measured by DopaFilm at the same location (*Figure 6—figure supplements 1–2*). These results suggest that Bassoon likely plays a key role in organizing dendritic release of dopamine in a manner that is reminiscent of its function in nerve terminals.

We similarly explored whether DopaFilm activity co-localizes with proteins involved in the exocytotic release of neurotransmitters. A myriad of cellular functions that require membrane fusion, including subcellular compartmentalization, cell growth, and chemical secretion, involve the use of SNARE-complex proteins (*Wickner and Schekman, 2008*; *Zhao et al., 2015*; *Weber et al., 1998*). SNARE proteins are responsible for triggering synaptic vesicle fusion as the last step of chemical neurotransmission (*Südhof, 2013*). Previous immunofluorescence studies have established that midbrain dopamine neurons likely use the vesicular SNARE protein synaptobrevin-2 (also known as VAMP2) as opposed to the more conventional variant synaptobrevin-1 (VAMP1) for exocytotic release

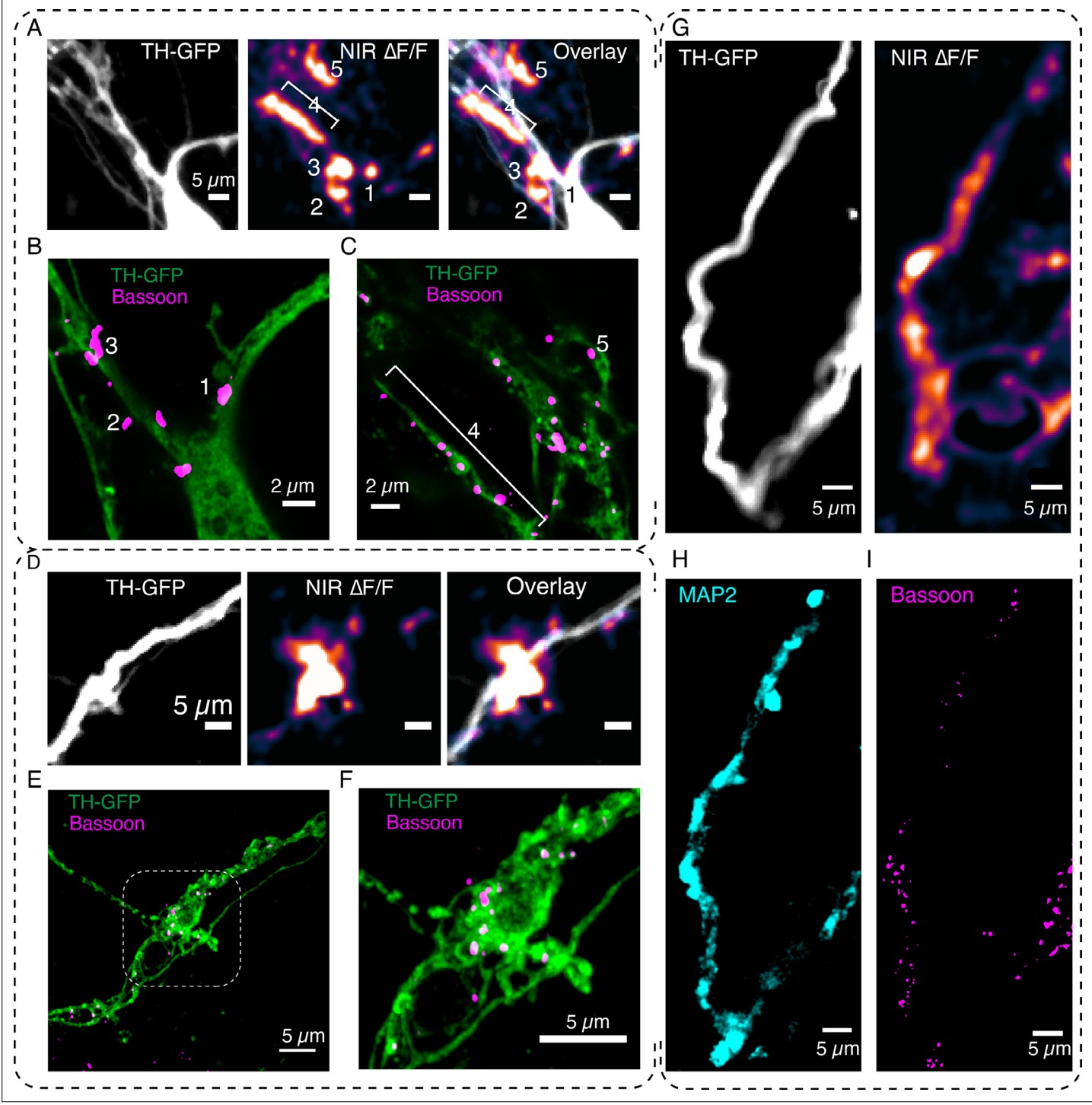

**Figure 6.** DopaFilm enables interrogation of the molecular correlates of dendritic release. (**A**) DopaFilm peak ΔF/F activity overlayed with tyrosine hydroxylase-GFP (TH-GFP) image around soma of dopamine neuron. Near-infrared (NIR) ΔF/F image contrast set at 0–10% ΔF/F. (**B, C**) Airyscan super-resolution images corresponding to field of view (FOV) in (**A**). (**D**) Dendrite of dopamine neuron and its ΔF/F activity. NIR ΔF/F image contrast set at 0–10% ΔF/F. (**E**) Airyscan super-resolution images corresponding to FOV in (**D**). (**F**) Close-up of boxed region depicted in (**E**). (**G–I**) Activity from a dendritic process and its corresponding TH-GFP, MAP2, and Bassoon images. The NIR ΔF/F image contrast was set at 0–30% ΔF/F.

The online version of this article includes the following figure supplement(s) for figure 6:

**Figure supplement 1.** Bassoon puncta density correlates positively with DopaFilm ΔF/F activity.

**Figure supplement 2.** Imaging activity from proximal dendrites associated with the soma of dopamine neurons.

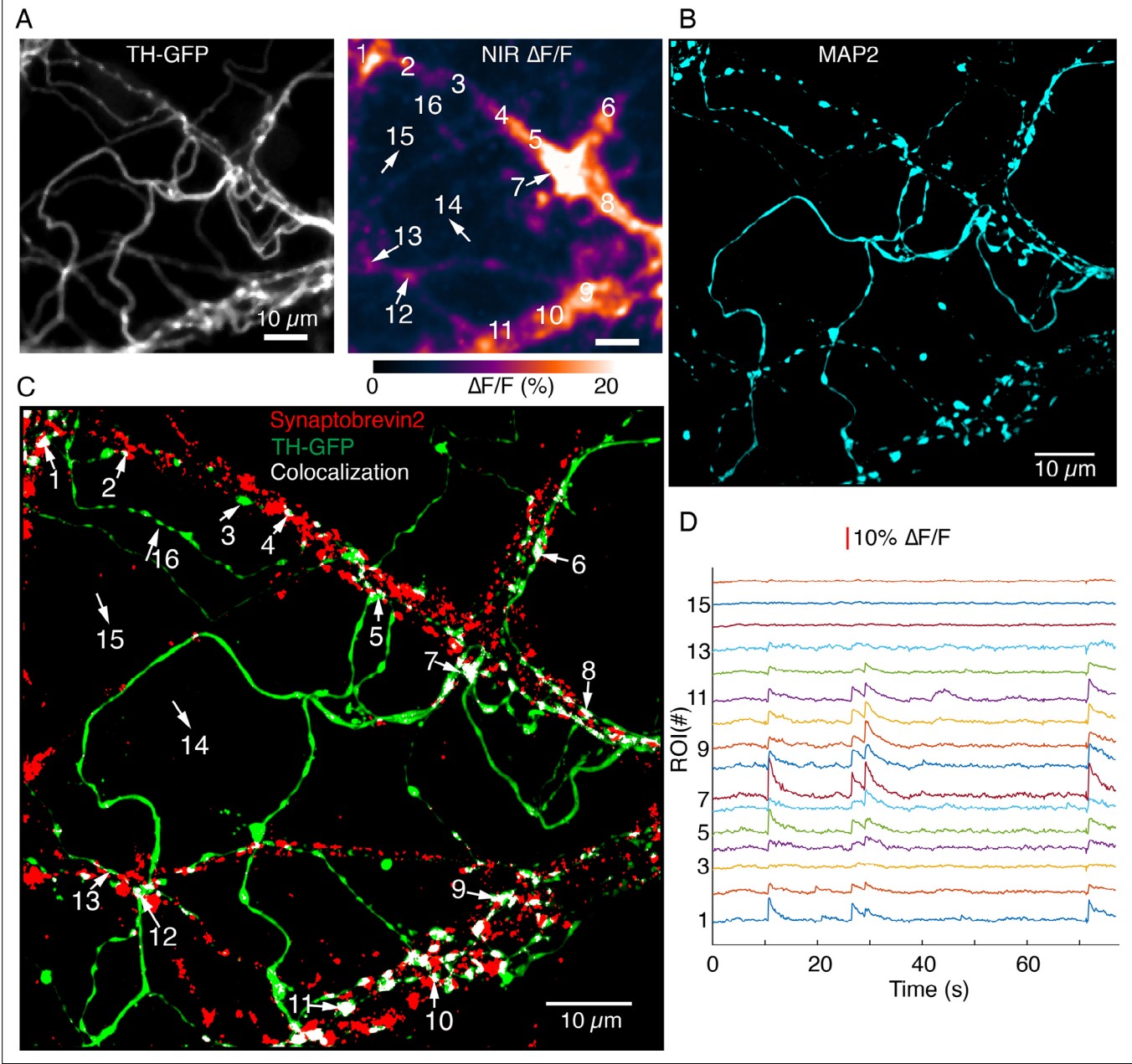

**Figure 7.** Synaptobrevin-2 is enriched at dendritic locations with high DopaFilm activity. (**A**) DopaFilm activity image in a dendritic arbor of autonomously spiking dopamine neuron (i.e. no external stimuli applied). Soma not shown. (**B**) MAP2 Airyscan image of tyrosine hydroxylase-GFP (TH-GFP) field of view in (**A**) shows the dendritic nature of processes in (**A**). (**C**) Airyscan image of TH-GFP image in (**A**) showing double-staining for TH-GFP (green) and synaptobrevin-2 (red). Co-localized green and red puncta are color thresholded to display white. Red puncta that do not co-localize with TH-GFP are from non-dopaminergic neurons in the co-culture system. (**D**) ΔF/F traces of regions of interest (ROIs) 1–15 depicted in (**A**) and (**C**). Notice that segments that have no synaptorevin-2 signal exhibit no DopaFilm activity (examples: ROIs 3, 14, 15, and 16).

(*Ludwig et al., 2016*; *Witkovsky et al., 2009*). We explored whether the expression of synaptobrevin-2 correlated with DopaFilm activity along a dendritic process. When imaging in dendritic arbors, we observed that DopaFilm activity along a dendritic process exhibited a strong correlation with the expression level of synaptobrevin-2, while TH+ segments with no synaptobrevin-2 expression lacked a corresponding DopaFilm activity (*Figure 7*).

## Discussion

Among design criteria for biological sensors is the requirement that activity be read out in temporal and spatial domains at sufficiently high sampling rates to recapitulate the underlying dynamics. From a temporal perspective, designing probes with suitable kinetics is carried out routinely during biosensor development. However, targeting the designed probes to read out the biochemical processes at appropriate spatial scales is not always trivial. As it pertains to chemical synapses, the nature of chemical effluxes demands uniform delivery of probes to the synapse and peri- and extra-synaptic spaces because such signals are inherently incompatible with cell-centered experimental paradigms. In this study, we introduced DopaFilm as a technology that facilitates visualization of chemical effluxes from dopamine neurons. DopaFilm inherently covers a large 2D plane and is therefore capable of detecting chemical efflux at the source (synapse) and tracks the signal to the end of its inherent spatial extent, subject to its limit of detection. We therefore circumvent the non-trivial demand of targeting the probe to the synapse and its surroundings. This design approach, coupled with the film's favorable temporal properties and quantal sensitivity, has facilitated visualization of synaptic dopamine chemical effluxes with unprecedented resolution. We used these capabilities to shed light on the nature of dopaminergic chemical synapses. We demonstrate that dopamine varicosities serve as beacons that broadcast dopamine release into neighboring peri- and extra-synaptic spaces. DopaFilm registered these release sites as hotspots of fluorescence activity and further afforded tracking of the evolution of the hotspots in the spatial domain, in real time.

To demonstrate the utility of DopaFilm, we applied it to study release in somatodendritic compartments of dopamine neurons. Dopamine release in midbrain regions, particularly in the substantia nigra pars compact (SNc) and ventral tegmental area (VTA), is a critical component of dopamine signaling for two important reasons. First, dopamine released in SNc and VTA is predicted to activate D2 auto-receptors, which in turn tune the excitability of dopamine neurons through GIRK channels (*Lacey et al., 1988*; *Ford, 2014*). This self-inhibition of dopamine firing in turn regulates dopamine release in distal regions that receive dense axonal innervation from midbrain dopamine neurons, including the striatum and prefrontal cortex, in addition to regulating release in SNc and VTA. Second, dendritic projections from the VTA into the substantia nigra pars reticulata (SNr) are thought to activate D1-receptors and regulate output from the primary GABAergic neurons of the SNr (*Ludwig et al., 2016*; *Miyazaki and Lacey, 1998*; *Radnikow and Misgeld, 1998*; *Threlfell and Cragg, 2007*). Through these two mechanisms, somatodendritic release plays a critical role in the wide range of functions that have been ascribed to dopamine neuromodulation, including motor control, motivation, and learning (*Ludwig et al., 2016*; *Ford, 2014*). Moreover, according to recent studies, somatodendritic release is posited to be responsible for delaying the onset of symptoms in Parkinson's disease, compensating for the extensive axonal degeneration that SNc dopamine neurons exhibit (*González-Rodríguez et al., 2021*). In sum, somatodendritic dopamine release is an important facet of dopaminergic neuromodulation and plays important roles in health and disease.

Despite this importance, however, insights into the spatiotemporal dynamics and regulatory mechanisms of somatodendritic release have been lacking, largely owing to the intermingling between soma and dendrites, the inability of available assays to tease out release at subcellular levels and the non-canonical nature of non-axonal release. We therefore deployed DopaFilm to study release in somatodendritic compartments of dopamine neurons. Our results show that somatodendritic release of dopamine arises primarily from the dendrites and not the soma. Major dendrites of dopamine neurons (i.e. the thick, smooth fluorescent trunks that emerge from the cell body, before ramification), as well as fine dendritic processes and dendritic arbors participate in dopamine release. Remarkably, dendrites with no apparent varicose morphologies participated in dopamine release as did dendritic arbors with axon terminal-like bouton structures. Dendritic activity shared much in common with dynamics observed in axons, but dendritic hotspots appeared localized in their spatial extent to the immediate vicinity of the release site in contrast to release in axonal processes, which propagated to a larger spatial extent. These differences may be driven by the dense axonal arbors of dopamine neurons, which give rise to higher number of release sites per unit area or could be driven by intrinsic differences between axonal and dendritic release sites. Dendritic release was observed to be robust, $Ca^{2+}$-dependent, and is broadcast from hotspots that are enriched with the presynaptic active zone protein Bassoon. Our study ascribes an important role for the SNARE-complex protein synaptobrevin-2 in dendritic dopamine release, consistent with predications from previous studies (*Witkovsky*

*et al., 2009*). Establishing the full complement of necessary and sufficient molecular machinery that organizes dendritic and axonal release will require systematic experimentation that will be the subject of future studies.

Although a rapidly growing suite of tools are being developed to facilitate measurement of neurochemical release, we believe that DopaFilm and other similar technologies will address a known need in neuroscience research. Most genetically encoded biosensors for neurochemicals are primarily optimized for large-scale multicellular imaging or fiber photometry experiments (*Patriarchi et al., 2018*; *Sun et al., 2018*; *Marvin et al., 2013*). In such applications, biosensors are broadly expressed on the plasma membrane of neurons, often in non-cell specific manner and are not targeted to synapses for single release site assay. Recent studies have sought to redeploy these biosensors to measure presynaptic release, but these approaches still rely on cell-surface anchoring strategies that limit their read out in spatial domain, and it remains to be established whether the sensors are truly presynaptically localized (*Dürst et al., 2019*; *Farsi et al., 2021*). Electrical or $Ca^{2+}$ recordings from postsynaptic processes can help us study presynaptic chemical synapses but these methods are lacking in throughput, represent summed synaptic input from multiple release sites, and are not suitable for neuromodulators that signal through G-protein coupled receptors (*Rosenmund et al., 1993*; *Hessler et al., 1993*; *Murphy et al., 1994*; *Oertner et al., 2002*; *Peled and Isacoff, 2011*). FM dyes, pHluorins, and other pH-sensitive vesicle-loadable optical tracers of synaptic vesicles require intense stimulation to drive the endo-exocytotic cycle and are not designed to visualize neurochemical diffusion at the requisite spatial scales (*Pereira et al., 2016*; *Ryan, 2001*; *Miesenböck et al., 1998*). Nevertheless, carefully designed experiments using pHluorins have enabled studies of presynaptic regulation of chemical synapses (*Balaji and Ryan, 2007*; *Maschi and Klyachko, 2017*; *Granseth et al., 2006*; *Gandhi and Stevens, 2003*). Redox active molecules like dopamine have benefited from amperometric measurements, which have reported quantal release events with submillisecond temporal resolution (*Hochstetler et al., 2000*; *Pothos et al., 1998*). Furthermore, GIRK-mediated postsynaptic currents have been used to measure quanta of dopamine release (*Gantz et al., 2013*). Here too, however, measurements are low throughput single point assays, and cannot to be localized to presynaptic release structures, and do not convey spatial information. In conclusion, up to now, measurements of neurochemical release without full spatiotemporal resolution and quantal sensitivity have limited our ability to visualize and understand synaptic chemical effluxes, to study the heterogeneity of chemical synapses within the same cell or across cell types, to investigate the complex regulatory mechanisms of presynaptic release, and to understand how these get shaped during synaptic plasticity, development, and disease. Therefore, new technologies that augment the capabilities of existing tools can make meaningful contributions to the study of neuroscience.

Synthetic strategies such as DopaFilm can fill a gap in the suite of tools that are designed to study cellular chemical release. Moreover, DopaFilm's non-photobleaching fluorescence in the NIR to SWIR regions of the electromagnetic spectrum (0.85–1.35 μm) is spectrally compatible with existing optical technologies, as we have demonstrated in this study. However, the unique spectral properties of DopaFilm means that microscopes that typically rely on silicon technology for photon detection (cameras and photomultiplier tubes) are not compatible with its use. DopaFilm requires use of detectors with InGaAs sensors for photon detection and additional optimization of optical components to facilitate transmission of NIR and SWIR photons. This requires an investment by the end user, albeit a modest one. Furthermore, single-wall carbon nanotube-based nanosensors, from which DopaFim is fabricated, have been developed for electroactive neurochemicals (catecholamines and indolamines; *Beyene et al., 2018*; *Jeong et al., 2019*) but strategies to sense molecules such as glutamate and GABA, and neuromodulators such as neuropeptides are still active areas of research. This would restrict DopaFilm-like strategies to the study of molecules for which we have robust nanosensors now. Finally, while synthetic probes may not have the benefit of familiarity that genetically encoded probes possess, they lend themselves to deployment in unique preparations such as one demonstrated in this study and ex vivo brain slices as previously demonstrated (*Beyene et al., 2019*). In the future, we believe that these synthetic constructs can offer opportunities to study dynamics of neurochemicals in less well-characterized regions, such as the retina and olfactory bulb for dopamine, as well as in non-model organisms that may not be compatible with existing genetic strategies.

# Materials and methods

## Key resources table

| Reagent type (species) or resource | Designation | Source or reference | Identifiers | Additional information |
|---|---|---|---|---|
| Strain, strain background (*Rattus norvegicus*) | Sprague Dawley | Charles River | Strain code: 001 | |
| Antibody | Anti-tyrosine hydroxylase (chicken polyclonal) | Aves Labs | TYH 6767979 | Dilution factor (1:1000) |
| Antibody | Anti-tyrosine hydroxylase (rabbit polyclonal) | Abcam | ab112 | Dilution factor (1:2000) |
| Antibody | Anti-tyrosine hydroxylase (mouse monoclonal) | Sigma-Aldrich | T2928 | Dilution factor (1:2000) |
| Antibody | Anti-Bassoon (mouse monoclonal) | Abcam | ab82958 | Dilution factor (1:2000) |
| Antibody | Anti-MAP2 (chicken polyclonal) | Novus Biologicals | NB300-213 | Dilution factor (1:1000) |
| Antibody | Anti-GFP (rabbit polyclonal) | Chromtek | PABG1 | Dilution factor (1:2000) |
| Antibody | Anti-synaptobrevin 2 (guinea pig monoclonal) | Synaptic Systems | SySy 104 318 | Dilution factor (1:1000) |
| Antibody | Brilliant Violet 421 AffiniPure (donkey anti-chicken polyclonal) | Jackson Immunoreseach Labs | 703-675-155 | Dilution factor (1:1000) |
| Antibody | Highly cross-adsorbed secondary antibody, Alexa Fluor 488 (goat anti-rabbit polyclonal) | Invitrogen | A11034 | Dilution factor (1:2000) |
| Antibody | Highly cross-adsorbed secondary antibody, Alexa Fluor 647 (goat anti-mouse polyclonal) | Invitrogen | A21236 | Dilution factor (1:2000) |
| Antibody | Highly cross-adsorbed secondary antibody, Alexa Fluor 647 (goat anti-guinea pig polyclonal) | Invitrogen | A21450 | Dilution factor (1:2000) |
| Antibody | Recombinant secondary antibody, Alexa Fluor Plus 647 (goat anti-rabbit polyclonal) | Invitrogen | A55055 | Dilution factor (1:2000) |
| Other | pAAV2.5-TH-GFP | Addgene | #80,336 | Adeno-associated virus (AAV) |
| Other | pAAV-Syn-ChrimsonR-tdT | Addgene | #59,171 | Adeno-associated virus (AAV) |
| Chemical compound, drug | APTES | Sigma-Aldrich | 440,140 | CAS: 919-30-2 |
| Chemical compound, drug | HiPco RAW single walled carbon nanotubes | Nanointegris | Batch #HR34-103 | |
| Sequence-based reagent | 5'-GTG TGT GTG TGT-3' oligonucleotide | Integrated DNA technologies | Standard Desalted | |
| Chemical compound, drug | Poly(D-lysine) | Sigma-Aldrich | P6407 | |
| Chemical compound, drug | NBQX disodium salt | Tocris | 1044 | CAS:479347-86-9 |
| Chemical compound, drug | Dopamine hydrochloride | Sigma-Aldrich | H8502 | CAS:62-31-7 |
| Chemical compound, drug | D-AP5 | Tocris | 0106 | CAS: 79055-68-8 |
| Chemical compound, drug | TTX | Tocris | 1078 | CAS: 4368-28-9 |
| Chemical compound, drug | Nomifensine maleate salt | Sigma-Aldrich | N1530 | CAS: 32795-47-4 |
| Chemical compound, drug | Triton X-100 | Millipore Sigma | T8787 | |
| Software, algorithm | MATLAB | R2020b | | |
| Software, algorithm | Fiji | 2.0.0-rc-59/1.51 k | See Ref 74 | |
| Software, algorithm | µManager | Version 1.4 | See Ref 73 | |

## Nanosensor synthesis and characterization

Single-stranded 5′-GTG TGT GTG TGT-3′ [(GT)$_6$] DNA oligonucleotides were purchased from Integrated DNA technologies (standard desalting and lyophilized powder), and HiPco single wall carbon

nanotubes were purchased from Nano Integris (batch #HR35-141). Solution-phase nanosensor synthesis was carried out by first mixing 1 mg of $(GT)_6$ and 1 mg of HiPco SWNT in 1 mL of 1 × PBS. Then the solution was bath-sonicated (Branson 1800) at room temperature for 20 min followed by probe-tip sonication (Sonics Vibra Cell) for 15 min in an ice-bath. Resulting suspension was centrifuged at 20,000 rcf (Eppendorf 5430 R) at 4°C for 60 min, and supernatant was carefully transferred into a new Eppendorf tube and stored at 4°C until further use. For characterization, each sensor batch was diluted to a working concentration of 10 ppm in 1 × PBS. Fluorescence and absorbance measurements were carried out on NS Super NanoSpectralyzer (Applied Nanotechnologies). Solution-phase fluorescence measurements and dopamine response tests were carried out on a custom built near infrared plate reader using flat clear bottom black 96-well plates (Corning, #3904). Solution-phase ΔF/F values were calculated from integrals of fluorescence counts between 875 and 1300 nm for pre- and post-dopamine fluorescence spectra.

## DopaFilm fabrication

Fabrication of DopaFilm on glass substrates was initiated by a silane-based surface modification reaction. First, 35 mm gridded dishes (MatTek P35G-1.5–14 C-GRID) were cleaned sequentially with 200-proof ethanol and copious amounts of molecular biology grade water. The cleaned dishes were incubated with 1 mL of 1% (3-aminopropyl) triethoxysilane (APTES; Sigma Aldrich) in ethanol for 1 hr at room temperature. After silane functionalization, dishes were rinsed three times with 2.5 mL of molecular biology grade water. Subsequently, 500 µL of 15 ppm nanosensor solution was added to the glass coverslip of the dish dropwise, allowing the nanosensor to spread evenly. After an overnight incubation at room temperature, excess nanosensor solution was aspirated and carefully rinsed with 2.5 mL of molecular biology grade water. To facilitate neuronal growth on DopaFilm, 1 mL of 0.05 mg/mL poly-D-lysine hydrobromide (PDL; Sigma Aldrich P6407) was applied on top and incubated at room temperature for 1 hr. Finally, dishes were thoroughly rinsed three times with 2.5 mL of molecular biology grade water and stored in sterile 1 × PBS until seeding neurons. Neurons were seeded directly on the engineered surface immediately after removing storage PBS solution followed by washing with sterile water to remove excess salts.

## Neuron co-culture on DopaFilm and viral infections

Primary rat neuronal culture work was conducted according to the Institutional Animal Care and Use Committee (IACUC) guidelines of Janelia Research Campus of the Howard Hughes Medical Institute. Sprague Dawley (Charles River Laboratories) neonatal rat pups were euthanized, cortical and hippocampal hemisphere tissue and SNc tissue dissected, and dissociated in papain enzyme (Worthington Biochemicals) in neural dissection solution (10 mM HEPES pH 7.4 in Hanks' balance salt solution) at 37°C water bath. After 30 min, enzyme solution was aspirated, and tissue pieces were subjected to trituration in 10% fetal bovine serum containing MEM media. Following trituration, cell suspension was filtered and resulting single-cell suspension was centrifuged to yield a cell pellet. Cell pellet was resuspended in Plating media (28 mM glucose, 2.4 mM $NaHCO_3$, 100 µg/mL transferrin, 25 µg/mL insulin, 2 mM L-glutamine, and 10% fetal bovine serum in MEM), and cell counts were recorded prior to seeding on DopaFilm substrate. To obtain healthy dopaminergic cultures, midbrain cells were seeded as co-cultures between SNc cells and cortical/hippocampal hemisphere cells at a ratio of 10:1 (~ 300 k cells per 35 mm dish). Initially, cells were seeded in attachment media (1:1 [v/v] plating media to NbActiv4 [BrainBits-NB4]), and cultures were maintained in a 5% $CO_2$ humid incubator at 37°C to facilitate cell adherence. After ~ 2 hr, attachment media was aspirated and growing media consisting of plating media: NbActiv4 at 1:20 ratio (v/v) was used. After the first week, cultures were fed twice a week by replacing old media 1:1 (v/v) with fresh NbActiv4 media. To express opsins and identify dopaminergic neurons, cultures were infected at 5 days in vitro (DIV) with titer-matched viruses using pAAV-Syn-ChrimsonR-tdT (Addgene #59171) and pAAV2.5-TH-GFP (Addgene #80336), respectively ($10^5$ combined infectious units per 1 mL of neuronal growth media). Plasmid DNA preparations and virus packaging were performed by Molecular Biology and Viral Tools facilities at Janelia Research Campus.

## Microscopy and imaging

For broad-spectrum (visible to SWIR, 400–1400 nm) imaging, we developed a custom microscope based on Thorlabs Bergamo microscope body. We modified the Bergamo to facilitate optimal

imaging in 400–1400 nm with widefield epifluorescence and confocal laser scanning modalities, using commercially available or custom-ordered optical components that maximize photon collection, transmission, and detection in the 400–1400 nm range. We used the widefield epifluorescence modality of the microscope for all our imaging experiments in this study. The microscope is equipped with two fiber-coupled NIR lasers to excite far red, NIR and SWIR fluorophores: a 671 nm laser (GEM 671, Laser Quantum) and a 785 nm laser (Thorlabs S4FC785). Additionally, the microscope is equipped with the following 4-wavelength LED light source (Thorlabs LED4D067) to excite and/ or actuate visible range fluorophores and opsins: 405, 470, 561, and 625 nm. The microscope is equipped with an InGaAs camera with optimized sensitivity in the SWIR range (Ninox 640 II, >85% QE in 1000–1500 nm, Raptor Photonics), and an sCMOS camera (CS2100M-USB, Thorlabs) for visible range imaging. SWIR images were acquired with μManager, an open source microscopy software (*Edelstein et al., 2014*).

Primary midbrain neuronal cultures were imaged 35–40 DIV. At the beginning of a typical imaging experiment, we replaced the NbActiv4 media with ACSF composed of, in mM: 124 NaCl, 2.5 KCl, 1.25 $NaH_2PO_4$, 24 $NaHCO_3$, 12.5 glucose, 5 HEPES, 2 $CaCl_2$, 2 $MgSO_4$, and spiked with 5 ppm of nanosensor solution for 1 hr. We then replaced the media with fresh ACSF and mounted the culture on the microscope for imaging. We used a 10× objective (Nikon N10XW-PF, 0.3 NA, and 3.5 mm WD) to identify a dopaminergic neuron from the co-culture system. We then switched to 40× objective (Nikon N40X-NIR, 0.8 NA, and 3.5 mm WD) to carry out activity imaging in the identified dopamine neuron. Before each imaging session, we record a GFP image of the TH signal in the green channel. We then switch to the NIR/SWIR channel for activity imaging. Imaging is carried out at frame rates of 10–20 frames per second depending on the brightness of the DopaFilm substrate (50–100 ms exposure time). If the neuron exhibits autonomous spiking activity, we do not apply any external stimuli. If the neuron does not exhibit spiking activity, we apply optical stimulus to evoke activity. Optical stimulation is driven through a custom-built MATLAB code that simultaneously controls and synchronizes the camera and optogenetic stimulation light source through external TTL pulses and time-locks stimulation times with specific camera frames. Optical stimuli applied to evoke activity are: 5 pulses, 25 Hz, 5 ms pulse width, and at power of 1 mW/mm². We use ChrimsonR to evoke activity and use 561 nm LED (Thorlabs LED4D067 with DC4100 driver) of the four-wavelength LED light source for stimulation. For repeat stimulation experiments, we used a rest period of ~ 2 min in between sessions. All imaging sessions were carried out at room temperature in ACSF. When imaging in $Ca^{2+}$-free media, we switched our imaging buffer to $Ca^{2+}$-free ACSF. When imaging in NOM or TTX, we spike the imaging ACSF with a known volume and concentration of a NOM or TTX solution to attain the target concentration of NOM or TTX. We incubate the culture for 10 min at room temperature before carrying out postdrug activity imaging.

## Electrophysiology

For electrophysiological recordings from hippocampal neurons, cultures were prepared as described above but with tissue extracted from the hippocampus only. Recordings were performed 14–16 days after plating at room temperature in the whole-cell voltage clamp and current clamp configurations using an Axopatch 200B amplifier (Molecular Devices). Data were acquired using pCLAMP 10 software and a Digidata 1440 A (Molecular Devices), filtered at 5 kHz, and digitized at 20 kHz. Cells were bathed in external solution containing (mM): 140 NaCl, 10 HEPES, 5 glucose, 4 KCl, 2 $CaCl_2$, 1 $MgCl_2$, and pH 7.4 adjusted with NaOH; osmolality 300 mOsm. Pipette internal solution contained (mM): 130 K-gluconate, 10 HEPES, 10 KCl, 1 EGTA, 4 Mg-ATP, 10 phosphocreatine, 0.3 Na-GTP, and pH 7.3 adjusted with KOH; osmolality 280 mOsm. Whole-cell patch pipettes had tip resistances of 3–7 MΩ and data were collected from cells with access resistance <25 MΩ. Membrane potentials were offline corrected for a –12 mV liquid junction potential. The electrophysiological parameters of neurons including capacitance, input resistance, and access resistance were determined in voltage clamp configuration by a 5 ms pulse from –80 mV to –70 mV. In current clamp, the resting membrane potential was recorded prior to current injection with series resistance compensated up to 80%. Rheobase firing threshold was measured by injecting cells with sequentially increasing +10 pA current injections of 100 ms duration. The minimum current required to evoke an action potential was defined as the rheobase. The firing rate, voltage threshold, and action potential properties were determined from a 1 s injection of current at ~ 2× rheobase. The minimum time between current injections under

each protocol was 5 s. Data are presented as mean ± SD, and statistical comparisons were performed by unpaired Mann-Whitney U tests.

## Data analysis

Each imaging stack was processed with a custom-built MATLAB code that converts the raw movie stack into a $\Delta F/F_0$ stack and generates correlated pixel components that we identify as hotspots of activity. To compute $\Delta F/F_0$, we first convolve the fluorescence time series movie (i.e. raw F values) with a 2D gaussian ($\sigma = 0.5$ pixels). To obtain $F_0$, a leaky cumulative minimum is calculated for each pixel in F, followed by repeated lowpass filtering to converge on a smooth $F_0$ that obeys the minima. $\Delta F$ is then calculated as $F–F_0$. Next, we partition the image into correlated components by applying non-negative matrix factorization (NNMF) to $\Delta F$ with sparsity and contiguity constraints. This partitions the image into largescale correlated regions (i.e. into pixel clusters with correlated activity). See *Figure 2—figure supplement 3*, *Figure 4—figure supplement 1*, *Figure 5—figure supplement 6*, e.g., NNMF components. For each NNMF component, $\Delta F/F_0$ is then calculated using the component's clustered pixels. Heatmaps and timeseries of $\Delta F/F_0$ traces are then generated for each movie file using native MATLAB capabilities. For simplicity, we refer to $\Delta F/F_0$ as $\Delta F/F$ in most of our figures and text. To calculate temporal FWHM and determine amplitude of $\Delta F/F$ traces, we used MATLAB's peak finder program. To calculate $\tau_{off}$, we fit first-order decay curves to identified local maxima. To determine $\tau_{peak}$, we used temporal difference between the location of the peak and the maxima of the second derivative of the trace. Statistical model fits and model discrimination tests were carried out using code written in MATLAB. All data analyses were done in MATLAB 2020 using the code located at the following repository: https://github.com/davidackerman/nnmf; *Ackerman, 2022* (copy archived at swh:1:rev:cff3d16a70b4fd4354e486347e38e7a9a1d167be). To compute centroids of hotspots and to generate overlay images, we used Fiji (*Schindelin et al., 2012*). To analyze Bassoon colocalization with $\Delta F/F$, we used Fiji. First, we aligned TH-GFP images taken on the NIR camera and the Airyscan PMT using image registration plugins and exploited TH-GFP image features as landmark to achieve this. We then registered the NIR $\Delta F/F$ peak frame onto the TH-GFP image that was aligned to the Airyscan image and collected $\Delta F/F$ and Bassoon intensity data at matched ROIs.

## Statistics

All statistical tests of significance (p-values) were computed and reported from Student's unpaired two-tailed t-test unless noted otherwise. We used MATLAB's built-in boxplot algorithm to display distributions for some of our data. The definitions for relevant features of the box plot are as follows: red line = median, edges of box: 25th and 75th percentile, top and bottom hash lines: minimum and maximum values of non-outlier data, and red points: outlier data.

## Post hoc immunocytochemistry and Airyscan super-resolution imaging

After activity imaging, cultures were fixed using 4% paraformaldehyde in 1 × PBS for 15 min at room temperature and washed twice with 1 × PBS. Next, cells were permeabilized/blocked in with 2% BSA in 1% Triton X-100 in 1 × PBS (PBST) for 1 hr at room temperature. All primary antibodies were incubated overnight at 4°C and secondary antibodies for 1 hr at room temperature in PBST containing 1% BSA. Primary antibodies used in this study include chicken anti-TH (1:1000; TYH 6767979, Aves labs), mouse anti-TH (1:2000; T2928, Millipore Sigma), rabbit anti-TH (1:2000; ab112, Abcam), mouse anti-Bassoon (1:2000; ab82958, Abcam), chicken anti-MAP2 (1:1000, NB300-213, Novus Biologicals), and guinea pig anti-synaptobrevin-2 (1:1000; SySy 104 318, Synaptic Systems). TH-GFP signal (after activity imaging) was amplified using a rabbit anti-GFP (1:2000; Chromotek PABG₁) antibody. Anti-mouse and anti-guinea pig secondary antibodies conjugated to Alexa Fluor 488 and Alexa Fluor plus 647-(Invitrogen) were applied at 1:2000 dilutions, and. anti-chicken secondary antibody conjugated to Brilliant Violet 421 (Jackson Immunoresearch laboratories) was added at 1:1000 dilution. Super-resolution microscopy was performed at the Janelia Imaging Facility on a Zeiss LSM 880 inverted confocal microscope equipped with a Plan-Apochromat 63×/1.4 NA oil objective in Airyscan mode using a hexagonally packed 32-channel gallium arsenide phosphide photomultiplier tube (120 nm lateral resolutions). Dopaminergic neurons of interest from the activity images were located by using MatTek dish grid location in the reflection mode.

## Materials availability

There are no known availability issues currently noted. DopaFilm is fabricated from commercially available reagents as detailed above. All other reagents used in this study are commercially available.

## Acknowledgements

This study was supported by the Howard Hughes Medical Institute through Janelia Research Campus (JRC). We thank Kristen Delevich, Joshua Dudman, Markita Landry, Luke Lavis, Timothy Ryan, Eric Schreiter, David Stern, and Linda Wilbrecht for discussions and comments on the manuscript. We express our gratitude to Dmitri Tsyboulski (Janelia Experimental Technology), and Wei Sun and Paulo Chaves (Thorlabs) for assistance with development of the visible-SWIR broad spectrum imaging microscope. We thank the Molecular Biology and Viral Core facilities at JRC for their assistance and vivarium team for support with animal husbandry. We thank Kaspar Podgorski for developing and sharing with us the MATLAB program used for data analysis and David Ackerman for improving the program to suit our needs. We thank Shu-Hsien Sheu and Boaz Mohar for sharing reagents and for helpful discussion on experiments. We are grateful for the support provided by imaging specialist Damien Alcor and the Advanced Imaging Center team at JRC. Ben Cristofori-Armstrong acknowledges the Australian National Health & Medical Research Council CJ Martin Fellowship APP1162427 for support.

## Additional information

### Funding

| Funder | Grant reference number | Author |
|---|---|---|
| Howard Hughes Medical Institute | | Chandima Bulumulla<br>Andrew T Krasley<br>Ben Cristofori-Armstrong<br>William C Valinsky<br>Deepika Walpita<br>David Ackerman<br>David E Clapham<br>Abraham G Beyene |
| Australian National Health and Medical Research Council CJ Martin Fellowship | APP1162427 | Ben Cristofori-Armstrong |

The funders had no role in study design, data collection and interpretation, or the decision to submit the work for publication.

### Author contributions

Chandima Bulumulla, Andrew T Krasley, Data curation, Formal analysis, Investigation, Validation, Writing – review and editing; Ben Cristofori-Armstrong, William C Valinsky, Formal analysis, Investigation, Resources; Deepika Walpita, Investigation, Resources; David Ackerman, Formal analysis; David E Clapham, Resources, Writing – review and editing; Abraham G Beyene, Conceptualization, Data curation, Formal analysis, Funding acquisition, Investigation, Methodology, Project administration, Resources, Supervision, Validation, Visualization, Writing – original draft, Writing – review and editing

### Author ORCIDs

Chandima Bulumulla (ID) http://orcid.org/0000-0003-0324-2008
William C Valinsky (ID) http://orcid.org/0000-0001-7736-9146
David Ackerman (ID) http://orcid.org/0000-0003-0172-6594
David E Clapham (ID) http://orcid.org/0000-0002-4459-9428
Abraham G Beyene (ID) http://orcid.org/0000-0003-3896-2144

### Ethics

Primary rat neuronal culture work was conducted according to the Institutional Animal Care and Use Committee (IACUC) guidelines of Janelia Research Campus of the Howard Hughes Medical Institute.

**Decision letter and Author response**
Decision letter https://doi.org/10.7554/eLife.78773.sa1
Author response https://doi.org/10.7554/eLife.78773.sa2

---

## Additional files

### Supplementary files
• MDAR checklist

### Data availability
All data generated or analyzed during this study are included in the manuscript, supporting file and uploadedvideo files. Source data for this study can be accessed from the following repositories: Figshare: https://figshare.com/articles/figure/DopaFilm_Source_Data/19416875 Computer code used in data analysis is available on GitHub at URL noted below: https://github.com/davidackerman/nnmf, (copy archived at swh:1:rev:cff3d16a70b4fd4354e486347e38e7a9a1d167be).

The following dataset was generated:

| Author(s) | Year | Dataset title | Dataset URL | Database and Identifier |
|---|---|---|---|---|
| Beyene A | 2022 | DopaFilm_Source_Data | https://doi.org/10.6084/m9.figshare.19416875.v1 | figshare, 10.6084/m9.figshare.19416875.v1 |

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
