## [Editor Report]

This is a very exciting study that presents a novel approach to examining dopamine release with spatial precision that is so far unrivaled. This manuscript is also important and timely in the field of biosensor development and of potential interest to neuroscientists who study neurochemical release. It introduces a synthetic nanofilm with high spatiotemporal resolution and quantal sensitivity to dopamine measurement. By utilizing this technology to visualize sub-cellular dopamine efflux, the work provides new insights into the spatiotemporal dynamics and protein machinery of somatodendritic dopamine release. The authors identify hotspots for DA release and also provide evidence for DA release in the presence of TTX, suggesting the occurrence of quantal release.

---

## [Decision Letter]

**Decision letter after peer review:**

Thank you for submitting your article "Visualizing Synaptic Dopamine Efflux with a 2D Composite Nanofilm" for consideration by *eLife*. Your article has been reviewed by 3 peer reviewers, and the evaluation has been overseen by a Reviewing Editor and Lu Chen as the Senior Editor. The following individual involved in the review of your submission has agreed to reveal their identity: Bianxiao Cui (Reviewer #3).

Essential revisions:

1) All reviewers agree that the key claims of the manuscript are well-supported by the data, therefore, no new experiments are required for the revision.

2) There are a few inconsistencies in the data presentation, and some aspects of data analysis could be improved. Please see the detailed comments/suggestions made by the reviewers below.

*Reviewer #2 (Recommendations for the authors):*

The following are suggestions for additional analyses that may make the presentation of the data more clear and more broadly useful.

Release probability. The stimulated DA signals here provide an excellent opportunity to examine release probability for a single release site. If there were many stimuli delivered during each experiment, the authors should consider reporting the probability of seeing release events in the axon and dendrites.

Quantal release – axons. Data presented in Figure 4 suggests detection of the quantal release of DA (ie, resulting from a single vesicle). Please quantify the frequency of release events in +TTX from a single release site and/or across release sites. Do 'quantal release' events have roughly the same kinetic profiles across imaging sites? Seems like this should be true if a single vesicle is being released.

Figure 4F shows a frequency histogram for dF/F amplitudes in TTX. Are the data values from the same release site or from multiple release sites? Also, is there a statistical method that may be used to demonstrate that the fits provided represent clearly distinct fits rather than a skewed distribution?

Did the authors test for quantal release from the dendrites in the presence of TTX? If so, please comment.

*Reviewer #3 (Recommendations for the authors):*

In this article, the authors monitored the spatiotemporal dynamics of dopamine release in dendritic processes of dopaminergic neurons using synthesized DopaFilm nanosensors. The strength of this work is included in the "Public Review" section and won't be repeated here. Overall, this is a high-quality study and the experiments were systematically carried out. I support the publication of this work with revisions. Below are some questions and comments that shall be addressed.

1. During the co-culture of dopaminergic neurons with hippocampal and cortical neurons, are the density well controlled such that these neurons mostly form single-layer on the DopaFilm such that all dopamine release measured is from the actual dopaminergic neurons in direct contact with the film?

2. In Figure 3, the authors showed the localized dopamine release after stimulation. Would the time interval between stimulation events affect the localized signal?

3. Following up on the previous point, the authors also showed some spots with no dopamine release. What is the percentage of these 'silent' sites compared to all observed dopamine release events?

4. The authors mentioned the dendritic release of dopamine is less diffusive than axon release. However, if the diffusive coefficient of dopamine is constant within the DopaFilm, then what is the biophysics of this observation? Would this have anything to do with, for example, the effective surface area/numbers/spatial distribution of the release sits? More comments are needed here.

5. The authors mentioned DopaFilm transients clearly localized to MAP2-positive and TH-GFP- processes, but later showed dopamine release is more directly related to Bassoon puncta. More clarification between dopamine release and these processes is needed.

---

## [Author Response]

Reviewer #2 (Recommendations for the authors):The following are suggestions for additional analyses that may make the presentation of the data more clear and more broadly useful.Release probability. The stimulated DA signals here provide an excellent opportunity to examine release probability for a single release site. If there were many stimuli delivered during each experiment, the authors should consider reporting the probability of seeing release events in the axon and dendrites.

We find this possibility intriguing and did consider it. However, we hesitated to calculate release probability because we are not certain about the number of action potentials that are evoked by our optical stimulation protocol, which likely evokes multiple action potentials. This is reflected in release probability estimates based on current data, which are very close to one (100%). Depolarization of dopamine neurons under electrophysiological control and simultaneous imaging from DopaFilm could permit evaluation of release probability with the resolution of a single bouton, which will permit examination of variance of synaptic release and estimation of intrinsic release probability under single spike conditions. Future studies from the lab will pursue this line of inquiry. For the current study, we will note that release from boutons appears to have a high probability of release based on our data that tracks time course of response across repeated simulations (see newly generated supplementary Figure 3—figure supplement 1A-B).

Changes to manuscript in response to this comment:

1. We added a phrase at the end of an existing sentence in the Results section:

“In axon terminals, we observed that repeat stimulations can be carried out with rest periods of ~2-3 minutes between stimuli, giving rise to a consistent set of DopaFilm hotspot activities, and suggesting that release-competent dopamine neuron boutons likely have a high probability of release (Figure 3A, Figure 3—figure supplement 1A-B).”

Quantal release – axons. Data presented in Figure 4 suggests detection of the quantal release of DA (ie, resulting from a single vesicle). Please quantify the frequency of release events in +TTX from a single release site and/or across release sites. Do 'quantal release' events have roughly the same kinetic profiles across imaging sites? Seems like this should be true if a single vesicle is being released.

We now quantify the frequency of TTX quantal events pooled across many release sites. To provide better context, we also quantified the frequency of spontaneous release events and provided a comparison of their frequency of occurrence with those of TTX quantal events. Here, by spontaneous release events, we mean those that are not generated by stimulus-evoked nor cell-autonomous spiking events. These events tend to be stochastic and temporally uncorrelated when they occur, so they are easy to identify. Their peak ∆F/F also tends to be smaller. We showed that these spontaneous events have amplitudes that are comparable to those of TTX quantal events but exhibited temporal characteristics that are different (presented in the new panel E of main Figure 4). Intriguingly, we find that TTX events occurred with nearly twice the frequency of these spontaneous events (Figure 4—figure supplement 1H-I). To show that TTX kinetics have “roughly the same kinetic profiles” compared to evoked release activities, we compute decay time constants (τ_off_) for TTX quantal events and compared those with τ_off_ from axons in the same field of view before TTX application (for evoked release activities). We created cumulative frequency distributions of τ_off_ values in TTX and without TTX (evoked activities) (Figure 4—figure supplement 1E-F). We defined a parameter, τ_10/90_ as the *range* of τ_off_ for values that fall in the 0.1 to 0.9 quantile window in the cumulative distribution curve. In axon terminals without TTX, we determined τ_10/90_ <inline-graphic mimetype="image" mime-subtype="png" xlink:href="media/image1.png" /> 1.95 s, whereas τ_10/90_ for TTX was 1.15 s, suggesting that quantal events have kinetic properties that are statistically less spread, consistent with the reviewer’s expectations.

These analyses are presented as panels in a newly generated supplementary Figure 4—figure supplement 1.

Changes to manuscript in response to this comment:

1. A new supplementary Figure 4—figure supplement 1 is generated as a supplement for main Figure 4. Relevant panels are Figure 4—figure supplement 1 E-I.

2. Discussions that accompany these newly generated figures are added in the main text. The main text now reads:

“As expected, bath application of 10 µM TTX abolished synchronous, evoked release of dopamine but stochastic, temporally uncorrelated fluorescence transients, which we refer to as spontaneous release events, persisted (Figure 4A, Figure 4—figure supplement 1, Video S2 before TTX, Video S3 after TTX). The spatial extent of DopaFilm fluorescence hotspots were diminished after application of TTX (Figure 4A, 4B, 4D, Figure 4—figure supplement 1A, Figure 4—figure supplement 2), and the peak amplitude of transients were smaller compared to evoked release (Figure 4E). TTX

∆F/F peaks were comparable to spontaneous activity peaks detected before TTX addition (Peak ∆F/F (%) (Mean ± SD): 5.3 ± 1.6 for spontaneous activity vs. 7.9 ± 4.6 for TTX) but exhibited different kinetic characteristics as measured by FHWM (s) (Mean ± SD: 1.5 ± 1.0 for spontaneous activity vs 0.74 ± 0.70 for TTX, *p* < 10^-4^ in unpaired ttest) (Figure 4E). The decay time constants (τ_off_) for TTX transients were smaller than those of evoked releases (Mean ± SD (s): 3.83 ± 0.8 vs. 1.52 ± 0.53, Figure 4—figure supplement 1E). To evaluate the statistical spread of the observed τ_off_ values, we created a cumulative frequency distribution of τ_off_ values with and without TTX (for evoked release) and defined a parameter, τ_10/90_, as the *range* of τ_off_ for values that fall in the 0.1 to 0.9 quantile window in the cumulative distribution curve (Figure 4—figure supplement 1F). In axon terminals without TTX (for evoked release), we determined that τ_10/90_ ≈ 1.95 s, whereas τ_10/90_ for TTX data was ≈ 1.15 s, suggesting that TTX events have kinetic properties that are more similar to each other. Interestingly, we observed that TTX transients occurred with a frequency of (Mean ± SD) 0.29 ± 0.24 s^-1^ per release site, which contrasted with a frequency of (Mean ± SD) 0.16 ± 0.13 s^-1^ per release site for spontaneous activities before TTX addition (Figure 4—figure supplement 1H-I).”

Figure 4F shows a frequency histogram for dF/F amplitudes in TTX. Are the data values from the same release site or from multiple release sites? Also, is there a statistical method that may be used to demonstrate that the fits provided represent clearly distinct fits rather than a skewed distribution?

We would like to thank the reviewer for this thoughtful question. We agree that a quantitative justification needs to be provided for the Gaussian mixture model that was used to fit the experimental data.

To begin, we would like to clarify that these TTX data are pooled from across multiple imaging sites and are not from a single site. We now make this clear in the manuscript. In our original presentation of the work, we relied on qualitative assessment of the data and our knowledge of quantal release processes to choose a three component Gaussian mixture model (GMM) as a statistical model to fit the experimental data. We now provide a justification for choosing three Gaussians from among a possible set of GMM selections. Akaike Information criterion (AIC) scores are often used to discriminate among statistical model fits for experimental data, with lower AIC values corresponding to statistical models that explain the most variation with the least number of model parameters. We calculated AIC scores for GMMs containing one (single Gaussian) to four Gaussians (that is, a four-Gaussian mixture model). AIC scores decreased as the number of Gaussians increased, hitting a minimum for three component GMM. A three component GMM had much lower AIC score than a two component GMM (see Figure 4—figure supplement 3C). On the other hand, a four component GMM had a higher AIC score than a three component one, suggesting overfitting. Therefore, we conclude that three component GMMs are better than the alternatives explored.

To explore the possibility that our experimental data can simply be explained by a skewed distribution instead of a GMM, we compared our three-component GMM with a lognormal distribution. Lognormal distributions are positively skewed and are frequently encountered in biological processes. A single lognormal distribution is a two-parameter statistical model (µ and σ) and would therefore be treated favorably in AIC scoring compared to an eight parameter three-component GMM (three pairs of µ and σ and two mixing probability parameters, π). This is because AIC scoring rewards a model’s predictive power, while penalizing the number of parameters used to dissuade overfitting. When we compare the best fitting lognormal with our chosen three component GMM, the GMM still returned a lower AIC score than a lognormal fit, suggesting that it explained experimental data variance better than a skewed distribution. In addition to AIC, we used a secondary model discrimination method called a Q-Q plot (quantile-quantile plot) to support our conclusion that GMM is a better statistical model for the experimental data. In conclusion, while we do not know if our chosen model is the best statistical model from all models that can be considered for this task, it appears to be clearly better than the few alternatives we have considered here.

Changes to manuscript in response to this comment:

1. We produced a new supplementary Figure 4—figure supplement 3 to go along with the main Figure 4.

2. We added a new paragraph to the Results section.

The new text now reads:

“We pooled peak ∆F/F values obtained from TTX hotspot traces across the field of view of imaging and generated a histogram distribution of the peak ∆F/F values (Figure 4— figure supplement 3A-B). Inspection of single traces and the pooled histogram data suggested that DopaFilm transients in TTX may represent quantal events of different sizes or integer multiples of the same quantal size. To rationalize our experimental observation, we explored the suitability of statistical model fits for our experimental data. We first considered the use of Gaussian mixture models (GMM), which are often employed in quantal analysis of synaptic transmission events. Using Akaike information criteria (AIC) as a discriminating score, we evaluated the relative qualities of GMMs composed of one to four components. We found that a three-component GMM whose mean parameters (µ) are µ_1_ = 5.7%, µ_2_ = 9.6%, µ_3_ = 15.1% ∆F/F offered the best fit (Figure 4F, Figure 4—figure supplement 3C). The spacing of the mean parameters, µ, suggested that DopaFilm fluorescence transients in TTX are likely driven by a quantized biological process where each upward deflection in DopaFilm fluorescence represented some integer multiple of a unitary fusion event. A four component GMM returned a higher AIC score than the three component AIC, suggesting overfitting (Figure 4—figure supplement 3C). We considered the possibility that our experimental data may simply be better described by a positively skewed distribution function instead of a GMM. To evaluate this possibility, we compared our chosen three-component GMM with a lognormal distribution. Lognormal distributions are positively skewed and are frequently encountered in biological processes.^35^ A single lognormal distribution is a two-parameter statistical model (µ and σ) and would therefore be treated favorably in AIC scoring compared to an eight parameter three-component GMM (three pairs of µ and σ and two mixing probability parameters, π). AIC rewards a model’s predictive power, while penalizing the number of parameters used. In this way, multiple models fitting the same data can be compared. Here, we saw that the GMM’s AIC score was better than that of the lognormal model (Figure 4—figure supplement 3D-E), suggesting that GMM explained the observed experimental variance better than a simple skewed distribution. In addition to AIC, we employed a graphical model discrimination method called Q-Q plot (quantile-quantile plot), which also supported our conclusion that GMM is a better statistical model for the experimental data (Figure 4—figure supplement 3F-G). In sum, our experimental results and statistical analysis demonstrate that DopaFilm transients that are measured in TTX represent quantal release events of dopamine and, to our knowledge, offers the only experimental observation of putative single-vesicle fusion events in which the released neurochemical can be visualized in both spatial and temporal domains.”

Did the authors test for quantal release from the dendrites in the presence of TTX? If so, please comment.

TTX data that we have so far collected in somatodendritic regions eliminates activity. However, we have not fully explored if TTX quantal events can be seen in dendrites and we’re unable to make a conclusion based on the data that we have. We do observe stochastic spontaneous events in dendrites, which we already state in the manuscript.

Changes to manuscript in response to this comment:

No changes were made in response to this comment.

Reviewer #3 (Recommendations for the authors):In this article, the authors monitored the spatiotemporal dynamics of dopamine release in dendritic processes of dopaminergic neurons using synthesized DopaFilm nanosensors. The strength of this work is included in the "Public Review" section and won't be repeated here. Overall, this is a high-quality study and the experiments were systematically carried out. I support the publication of this work with revisions. Below are some questions and comments that shall be addressed.1. During the co-culture of dopaminergic neurons with hippocampal and cortical neurons, are the density well controlled such that these neurons mostly form single-layer on the DopaFilm such that all dopamine release measured is from the actual dopaminergic neurons in direct contact with the film?

We would like to thank the reviewer for this important comment. Controlling the density of neurons in dish is very important in the protocol. While we did note this importance in the original text, we acknowledge that is should be emphasized more.

Changes to manuscript in response to this comment:

1. We now have the following sentences in the manuscript:

“During seeding, we optimized density of cells on DopaFilm such that mature neurons formed a monolayer of cells on the substrate. Optimized seeding density allowed us to record activities arising from isolated dopamine neurons where no other neurons in the vicinity of the neuron of interest were TH^+^, ensuring that detected activity can be assigned to single identifiable processes with minimal crosstalk.”

2. In Figure 3, the authors showed the localized dopamine release after stimulation. Would the time interval between stimulation events affect the localized signal?

We addressed this point as a response to comments from reviewer 2. We will note that for multiple simulations applied within one imaging session, some short-term depression can be seen (see Figure 4C, left panel). However, with rest periods of 2-3 minutes between imaging sessions, we see a robust response across stimulations (see Figure 3—figure supplement 1A-B).

Changes to manuscript in response to this comment:

1. New Figure 3—figure supplement 1A-B addresses this comment.

3. Following up on the previous point, the authors also showed some spots with no dopamine release. What is the percentage of these 'silent' sites compared to all observed dopamine release events?

This is the same comment (#1 above) that is repeated here. It has been addressed.

4. The authors mentioned the dendritic release of dopamine is less diffusive than axon release. However, if the diffusive coefficient of dopamine is constant within the DopaFilm, then what is the biophysics of this observation? Would this have anything to do with, for example, the effective surface area/numbers/spatial distribution of the release sits? More comments are needed here.

We thank the reviewer for this interesting question, and we agree that some discussion related to this is warranted. We think that these differences in spatial propagation of dopamine released in axons and dendrites are likely driven by biological differences. One key difference is the density of release sites. Axonal arbors of dopamine neurons are highly ramified and produce a large number of release sites (boutons) per unit area. On the other hand, dendritic processes tend to be less dense and their ramifications are not as elaborate. Therefore, there is a notable difference in the “release site density” that explains some of these differences. Additionally, even on a per-release site bases, axon terminals appear to be more release capable per stimulation (see Figure 5—figure supplement 4C for peak ∆F/F comparisons). It is possible that more release-ready vesicles are available in axons than dendrites. The nature of vesicles in dendrites of dopamine neurons is not well understood and it will be interesting to examine dendritic release site ultrastructure with electron microscopy and compare that with axons. These will be explored in future studies.

Changes to manuscript in response to this comment:

1. We added the following statement in the Discussion section:

“These differences may be driven by the dense axonal arbors of dopamine neurons, which give rise to higher number of release sites per unit area or could be driven by intrinsic differences between axonal and dendritic release sites.”

5. The authors mentioned DopaFilm transients clearly localized to MAP2-positive and TH-GFP- processes, but later showed dopamine release is more directly related to Bassoon puncta. More clarification between dopamine release and these processes is needed.

We regret this potential source of confusion. MAP2 is a protein that we used to distinguish between axons and dendrites. MAP2 is absent in axons but it is enriched in dendrites. It is generally expressed throughout the cytoplasm of dendritic processes. On the hand, Bassoon is a protein that is selectively enriched at neurotransmitter release sites. Bassoon labeling is punctate. Therefore, it is possible to have both MAP2 and Bassoon signals at a given location. In our study, we used MAP2 to confirm that a neuronal process is dendritic, and then looked for Bassoon puncta in areas where we see hotspots of activity.

Changes to manuscript in response to this comment:

1. We now add the following clarifying sentence in the Results section:

“MAP2 is enriched in dendrites and is absent in axons ^40^; therefore, MAP2 immunoreactivity was used to distinguish between the dendritic vs. axonal nature of observed transients.”